# Cortical-limbic circuit dynamics of approach-avoidance conflict in humans

Brooke R. Staveland [1,2] ✉, Julia Oberschulte[3], Barbara Berger [1,2,4], Tamas Minarik[1,2], Olivia Kim-McManus [5,6], Jon T. Willie [7,8], Peter Brunner [7,8], Mohammad Dastjerdi [9], Jack J. Lin [10,11], Elizabeth L. Johnson [12], Michelle Paff [13], Ming Hsu [1,14,15] & Robert T. Knight [1,2,15]

Choosing to approach or avoid is common in everyday life and excessive avoidance is a cardinal feature of anxiety disorders. We use intracranial EEG to define a prefrontal-limbic circuit supporting approach and avoidance. Pre-surgical epilepsy patients (n = 20) performed an approach-avoidance conflict decision-making task inspired by the arcade game Pac-Man, where patients trade off rewards against losses from ghost attack. During approach, theta power increases across a limbic circuit including the hippocampus, amygdala, orbitofrontal cortex and anterior cingulate cortex, which drops during avoidance. Theta connectivity between this circuit and lateral prefrontal cortex increases during approach and falls during avoidance. Network connectivity tracks how long patients approach, with enhanced synchronicity extending approach times. During imminent threat, the system switches to sustained increase in high-frequency activity in the lateral prefrontal cortex. The results provide evidence of a distributed prefrontal-limbic circuit, mediated by theta oscillations and high frequency activity, underlying approach-avoidance conflict in humans.

Anxiety is a complex response to ambiguous and potentially negative future states. Anxiety comprises threat appraisal and uncertainty management, among other constituents, and dysfunction in any of these domains could result in persistent or pathological anxiety. A key behavioral paradigm for investigating both clinical and normative anxiety is approach-avoidance conflict. Approach-avoidance conflict describes the daily situations where a single choice entails potential gains and losses simultaneously, which reliably induces anxiety in everyday life[1,2]. For example, consider the decision to approach someone new while attending a party. The same action could result in the positive outcome of a nice conversation or a new friend, or a negative outcome such as social rejection or embarrassment. While anxiety in this context is normative, when avoidance becomes excessive (never speaking to anyone new, or avoiding social gatherings altogether) it becomes maladaptive[1,3,4]. Approach-avoidance decisions are dysregulated in generalized anxiety disorder (GAD), post-traumatic stress disorder (PTSD), and agoraphobia. Specifically, in clinical disorders there is maladaptive avoidance of potentially

[1]Helen Wills Neuroscience Institute, University of California, Berkeley, CA, USA. [2]Departments of Psychology and Neuroscience, University of California, Berkeley, CA, USA. [3]Department of Psychology, Harvard University, Cambridge, MA, USA. [4]School of Psychological Sciences, Victoria University of Wellington, Wellington, New Zealand. [5]Department of Neurosciences, University of California, San Diego, CA, USA. [6]Division of Neurology, Rady Children's Hospital, San Diego, CA, USA. [7]Department of Neurosurgery, Washington University School of Medicine, St. Louis, MO, USA. [8]National Center for Adaptive Neurotechnologies, Albany, NY, USA. [9]Department of Neurology, Loma Linda University, Loma Linda, CA, USA. [10]Department of Neurology, University of California, Davis, CA, USA. [11]Center for Mind and Brain, University of California, Davis, CA, USA. [12]Departments of Medical Social Sciences, Pediatrics, and Psychology, Northwestern University, Evanston, IL, USA. [13]Department of Neurological Surgery, University of California, Irvine, CA, USA. [14]Haas School of Business, University of California, Berkeley, CA, USA. [15]These authors jointly supervised this work: Ming Hsu, Robert T. Knight. ✉e-mail: brooke.staveland@ucsf.edu

aversive cues at the cost of large rewards[2,5,6], and this can be disambiguated from adaptive avoidance, such as avoiding cues that are likely to result in large costs, with low probability of reward (e.g., avoiding speaking to a known rude or unkind person at a party). Understanding the cortical and limbic neural circuits that regulate healthy and adaptive approach-avoidance conflict is key to understanding the neural underpinnings of both normal and excessive clinical anxiety.

There is a rich history defining the limbic system, long known to underlie aspects of emotional behavior[7–9], including approach-avoidance[10–12]. In 1937, Papez described brain regions, particularly the hippocampus and anterior cingulate, associated with emotional control[13] and coined the term the limbic system based on Broca's anatomical description of the 'limbic lobe' encircling the corpus callosum[14]. MacLean added the amygdala to the limbic system in response to reports of abnormal behavior in monkeys with bilateral amygdala lesions[15,16]. Based on the disordered emotional and social behavior of patient Phineas Gage[17], the orbitofrontal cortex completed the modern definition of the limbic system. Given the complexity and extent of the limbic system, we interrogated the role of all these regions with a focus on theta oscillations known to be involved in network communication in a host of human behaviors. We further hypothesized that the engagement of the lateral prefrontal cortex, specifically the middle frontal gyrus (MFG), in approach-avoidance would be distinct from limbic regions[2,18]. The MFG has been associated with cognitive control and emotion regulation, and nonhuman primates (NHPs) research has reported evidence that the prefrontal cortex is part of a cognitive circuit exerting top-down control on the limbic circuit[19]. Here, we use intracranial EEG (iEEG) to delineate a prefrontal-limbic system engaged in approach-avoidance including the middle frontal gyrus (MFG), orbitofrontal cortex (OFC), amygdala, anterior cingulate cortex (ACC) and hippocampus (HC) in humans.

Elements of this network have been explored in both rodents, nonhuman primates, and humans. Approach-avoidance conflict tasks, such as the elevated plus maze or the open field test have been used to study the neural circuits underlying anxiety in rodents[2,20,21]. In these tasks, there is a conflict between the positive drive to explore new areas versus the negative drive to avoid exposed areas. Using a variety of approaches, researchers have identified pairs of prefrontal-limbic regions, mediated by theta-band power, associated with avoidance behavior on these tasks[22–24]. Researchers have also found that a subset of ventral hippocampal (vHC) neurons projecting to medial PFC (mPFC) show increased theta synchrony in anxiogenic contexts[25]. Optogenetically stimulating these neurons at 4–12 Hz increases both synchrony between the vHC and the mPFC and increases avoidance behavior in rodents[24]. Similarly, optical activation of rodent basolateral amygdala neurons that terminate in the mPFC increases avoidance in the elevated plus maze, open field test, and other anxiogenic tasks, while inhibition of these neurons increases approach behavior[26]. Other studies report bidirectional mPFC-amygdala circuitry, implicating the mPFC in top-down control of amygdala-mediated anxiety behaviors in rodents[21,25]. However, the human homologue of the rodent mPFC is still debated[27] and the massive expansion of human prefrontal cortex limits a direct translation of rodent to human findings.

Similar regions are implicated in human approach-avoidance using non-invasive imaging and neuropsychology. Bilateral hippocampal lesioned patients showed increased approach behavior[28], and a magnetoencephalography study found that hippocampal-PFC theta synchrony increased with threat[29]. Some fMRI studies of human approach-avoidance conflict report activations in the amygdala[30], but others do not[31–33]. It is hypothesized that the amygdala responds equally to appetitive and aversive cues and may be missed in standard fMRI contrasts[2]. Other studies using fMRI have found activations in orbitofrontal cortex, anterior cingulate, and dorsolateral prefrontal cortex that correlate with task parameters like increasing threat,

reward, and conflict, though there is variation between different tasks[2,32–35]. Separately, researchers report that EEG midline frontal theta increases with approach-avoidance conflict, similar to midline frontal theta elevations in tasks requiring cognitive control[36]. Together, this work suggests that both prefrontal and limbic regions are engaged in approach-avoidance behavior. However, the real-time circuit dynamics of approach-avoidance have yet to be characterized in humans.

Here we test for the existence of such a prefrontal-limbic network underlying approach-avoidance by examining (1) task-dependent modulations in theta power in middle frontal gyrus and limbic regions including the hippocampus, amygdala, OFC and anterior cingulate; (2) elevated theta band coherence across these regions; and (3) dynamic changes in this circuit in response to increases in threat and reward. To test these hypotheses, we collected intracranial electrode recordings across this circuit in twenty epileptic patients during pre-surgical assessment while they performed a continuous-choice, approach-avoidance conflict task based on the arcade game Pac-Man.

In this work, we find that theta power is elevated during the approach period compared to the avoidance period in each of the limbic regions, but not in the MFG. Theta connectivity between this limbic circuit and lateral prefrontal cortex increases during approach and falls during avoidance. Furthermore, theta synchrony across limbic regions and MFG correlates with how long participants approach in a given trial. Finally, we find that high frequency activity in the middle frontal gyrus tracks threat when the participant is under direct attack by the Ghost.

## Results
### Behavioral Results

We designed an approach-avoidance conflict task based on the arcade game Pac-Man (See Fig. 1a). Briefly, on each conflict trial (80% of trials), Pac-Man, the Ghost, and dots are placed along a single corridor that runs horizontally across the screen. The participant controls the Pac-Man, and they are instructed to collect as many dots as they can while avoiding the Ghost, which is moving back and forth at a constant speed at the opposite side of the corridor. The participant makes a single button press, using the left or right arrow keys, to initiate movement along the corridor at a constant speed. By moving towards the center of the corridor, they can collect up to 5 dots, resulting in points, which varied in either small (10 points) or large (20 points) sizes and are placed in pseudorandom configurations between Pac-Man and the Ghost. Importantly, at any point in the trial, they could also choose to move away from the center of the corridor (and, thus, away from both the dots and the Ghost), where they could exit to the next trial at the end of the corridor. Critically, the distance between the Pac-Man and the Ghost directly determined the probability the Ghost would initiate an attack. During an Attack, if Pac-Man was "caught" by the Ghost, they lost one of their three lives and all the points acquired on that trial. Please see the Methods for the full description of the task. 20% of trials were conflict-free, where the participant was free to collect the reward without threat of the Ghost.

We validated that the task was inducing approach-avoidance trade-offs by assessing the behavior in a sample of 191 participants recruited from the online platform Prolific (see "Methods" for details). Across the twelve blocks of twenty trials each, participants collected an average of 3.83+/0.34 dots per trial, demonstrating that they often left some potential reward in the trial (max 5 dots) to maintain a greater distance from the ghost (See Fig. 1b). Participants on average turned away from the ghost when they were 35.3+/3.4 units away from the ghost (corresponding to ~17% of the length of the corridor), though there was across-subject variability in how close participants were willing to get to the ghost to receive more reward (See Fig. 1c). We also hypothesized that participants would make risk/reward trade-offs tolerating more risk to get large rewards. We tested this by running a

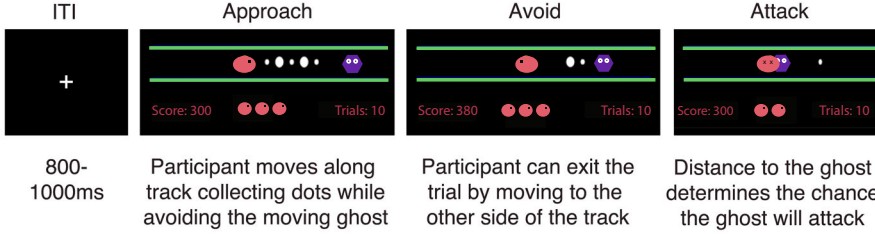

**a.** Task design for a continous-choice, approach-avoidance conflict task

| ITI | Approach | Avoid | Attack |
|---|---|---|---|
| 800–1000ms | Participant moves along track collecting dots while avoiding the moving ghost | Participant can exit the trial by moving to the other side of the track | Distance to the ghost determines the chance the ghost will attack |

**b-e** Task validation & behavioral data from iEEG and online sample of 191 particpants

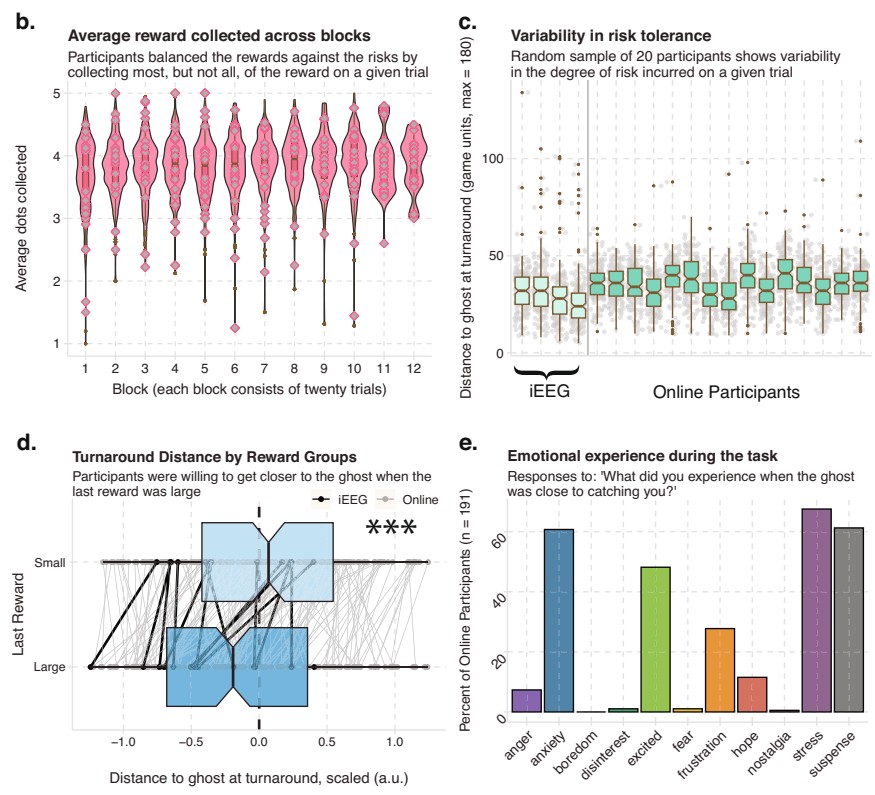

**b.** Average reward collected across blocks
Participants balanced the rewards against the risks by collecting most, but not all, of the reward on a given trial

**c.** Variability in risk tolerance
Random sample of 20 participants shows variability in the degree of risk incurred on a given trial

**d.** Turnaround Distance by Reward Groups
Participants were willing to get closer to the ghost when the last reward was large

**e.** Emotional experience during the task
Responses to: 'What did you experience when the ghost was close to catching you?'

**f.** 631 contacts from 20 participants were placed in prefrontal and limbic regions

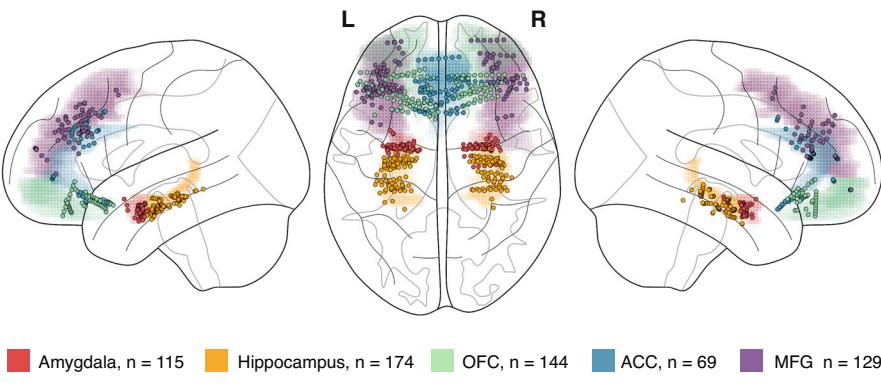

Amygdala, n = 115    Hippocampus, n = 174    OFC, n = 144    ACC, n = 69    MFG n = 129

linear mixed effects model, where a two-level factor specifying if the last reward was large or small was used to predict the distance to the ghost at the time of the decision to avoid. As predicted, when the last reward in the corridor was large, participants were willing to move closer to the ghost to collect the large reward (Estimate$_{small>large}$: 0.11, 95% CI [0.09–0.13], $P+ > 0.99$, BF$_{10}$ > 100, ESS$_{bulk}$ = 14861.4; see Fig. 1d, see Table 1 for model details; see Source Data File). We did not find

evidence that the self-reported sex of the participant altered how close participants were willing to get to the ghost (Estimate$_{male>female}$: −0.04, 95% CI [−0.16 to 0.08], ESS$_{bulk}$ = 1052), nor had any interaction with reward size (Estimate$_{SmallReward:Male}$: 0.03, 95% CI [−0.02 to 0.18], ESS$_{bulk}$ = 15356). Since approach-avoidance conflict is often used as a behavioral model of anxiety, we asked participants to report their emotions during the trial. Participants were given a set of emotions

**Fig. 1 | Task design, behavioral data and electrode placement. a** Task design for a real-time, approach-avoidance conflict task. Pac-Man (pink oval) was placed along a horizontal corridor, with a ghost (purple hexagon) moving in a set path back and forth at the opposite end of the corridor. Five rewards of varying sizes were placed between Pac-Man and the ghost. Participants used the left and right arrow keys to move closer to the ghost to collect reward (approach), or to move back to the end of the corridor where participants could exit the trial (avoid). In the task, participants viewed alternate graphics inspired by the arcade game Pac-Man. **b**–**e** Normative behavior on the task in iEEG patients and an online sample of 191 participants. **b** Average reward collected across blocks. Violin and box plots of the average number of rewards (dots) collected on each trial across 12 blocks of 20 trials. Box plots depict median and interquartile range across participants (*n* = 201), with whiskers covering most extreme values except outliers. Grey diamonds indicate the performance of each intracranial participant. Variability in risk tolerance. **c** Box plots for 20 example participants and 4 intracranial participants of the average distance to the ghost at the final decision to avoid across all trials. Box plots depict median and interquartile range across participants (n = 201), with whiskers covering most extreme values except outliers. Grey dots indicate the turnaround distance for each trial. **d** Turnaround distance by reward group. Box plots depict median and interquartile range across participants (*n* = 201), with whiskers covering most extreme values except outliers. Rewards could be large (worth 20 points) or small (worth 10 points). On trials where the last dot was large, participants moved closer to the ghost compared to trials where the last reward was small. Black dots indicate average distance for each intracranial patient, grey dots indicate the average for each online participant, and the lines link the behavior within participants. * indicates the 95% confidence intervals for the effect of reward size on turning distance did not include 0, in a Bayesian linear mixed effects model. **e** Emotional experience of the task in the online sample. *Y*-axis is the percent of participants who reported experiencing the emotion on the *X*-axis. Participants were allowed to select more than one emotional experience or write their own. **f** Electrode placement. 631 electrodes were placed across the prefrontal and limbic regions in 20 intracranial participants. Colored shading indicates region, and dots indicate electrode.

(anger, anxiety, boredom, disinterest, excitement, frustration, hope, stress, and suspense) as well as an option to write their own. Participants could report multiple emotions. A majority of participants reported experiencing anxiety, stress, and suspense during the task, supporting face-validity of the task (Fig. 1e).

Twenty patients with seizure- and artifact-free intracranial electrodes in the amygdala, hippocampus, OFC, ACC, and MFG (Fig. 1f; see methods and supplementary data for patient details) performed the task. Behavioral performance across the twenty patients was comparable to the performance in our online sample (See Fig. 1b–e). Electrodes near seizure foci were removed, and epochs where seizure activity spread beyond the local seizure onset zone were excluded. This left a total of 115 amygdala contacts, 174 hippocampal contacts, 144 OFC contacts, 69 anterior cingulate contacts, and 129 MFG contacts that were included in our analyses. While some patients did not have contacts in all five regions, there was always at least 12 patients with coverage in each possible *pair* of our five regions, allowing us to probe the neural dynamics across all sets of regions (See Supplementary Fig. 1 and Supplementary Table 1 for individual patient electrode coverage and demographics).

## Theta power is locally modulated by approach-avoidance decisions in limbic regions

We hypothesized there would be task-dependent theta power modulations within our regions of interest. In each region, we calculated time-frequency plots time-locked to the patients' choice to stop approaching and turn back to exit the trial. If the participant turned multiple times, we selected the final timepoint as the decision to avoid. We observed a similar pattern in the hippocampus, amygdala, OFC, and ACC, where low-frequency power (between 3–8 Hz) is elevated during the approach period of the trial but drops after the decision to avoid and exit the trial (See Fig. 2a).

We tested if the approach period correlated with greater theta power compared to the avoidance period across the different regions. We excluded the initial period of the trial where the Pac-Man was stationary to best compare the approach activity to avoidance activity. On average, patients began moving 1.02+/−0.46 seconds after trial onset and approached both the threat (Ghost) and reward (dots) for 1.39+/−0.22 s. We therefore averaged theta power during the final 1.5 s of approach and first 1.5 s of avoidance in each electrode and patient, on trials without a Ghost Attack. We estimated a Bayesian mixed effects model of theta power using their approach/avoidance decisions with hierarchically grouped random effects of patient and electrode (See Methods and Table 1 for details). We found that theta power was elevated during approach compared to avoidance in conflict trials within the hippocampus (See Fig. 2a; Conflict-Trial Estimate$_{approach>avoidance}$: 0.24, 95% CI [0.10-0.37], P + > 0.99, BF$_{10}$ > 4, ESS$_{bulk}$ = 3838.9, Ratio of Sig. Electrodes$_{P+ > 0.95}$ = 72/115), amygdala (Conflict-Trial Estimate$_{approach>avoidance}$: 0.16, 95% CI [0.07-0.26], P + > 0.99, BF$_{10}$ > 3, ESS$_{bulk}$ = 5903.8, Ratio of Sig. Electrodes$_{P+ > 0.95}$ = 59/101), OFC (Conflict-Trial Estimate$_{approach>avoidance}$: 0.21, 95% CI [0.13–0.29], P + > 0.99, BF$_{10}$ > 100, ESS$_{bulk}$ = 4243.2, Ratio of Sig. Electrodes$_{P+ > 0.95}$ = 127/167) and ACC (Conflict-Trial Estimate$_{approach>avoidance}$: 0.26, 95% CI [0.14-0.38], *P* + > 0.99, BF$_{10}$ > 18, ESS$_{bulk}$ = 7222.9, Ratio of Sig. Electrodes$_{P+ > 0.95}$ = 50/73). In the subset of control trials that were conflict-free, i.e., where patients collected reward without threat of the ghost, we ran an equivalent model, where the approach period (e.g., 1.5 s before the decision to turn around, usually made after all rewards were collected) was compared against the 1.5 seconds after the decision to turn. We did not find a significant difference in theta power between the approach and return windows (Hippocampus Estimate$_{approach>return}$: 0.08, 95% CI [0.08–0.25], *P* += 0.84, BF$_{10}$ = 0.07, ESS$_{bulk}$ = 2778.1, Ratio of Sig. Electrodes$_{P+ > 0.95}$ = 72/115; Amygdala Estimate$_{approach>return}$: 0.11, 95% CI [−0.02 to 0.22], P += 0.96, BF$_{10}$ = 0.14, ESS$_{bulk}$ = 4697.1, Ratio of Sig. Electrodes$_{P+ > 0.95}$ = 38/101; OFC Estimate$_{approach>return}$: 0.13, 95% CI [−0.01 to 0.27], P += 0.97, BF$_{10}$ > 0.21, ESS$_{bulk}$ = 4350.9, Ratio of Sig. Electrodes$_{P+ > 0.95}$ = 71/167; ACC Estimate$_{approach>return}$: 0.12, 95% CI [−0.07 to 0.31], P += 0.90, BF$_{10}$ > 0.11, ESS$_{bulk}$ = 7437.9, Ratio of Sig. Electrodes$_{P+ > 0.95}$ = 24/73), indicating that the effects were not likely driven by purely spatial or reward-only behavior (See Supplementary Fig. 2 for TFRs of the conflict-free trials, see Source Data File for model estimates).

We then tested for theta power modulations in the MFG, implicated in previous fMRI and nonhuman-primate studies of approach-avoidance but not considered to be part of the limbic system[18,19,34]. We did not find evidence for increased theta power in the approach period compared to the avoidance period in the MFG (Conflict-Trial Estimate$_{approach>avoidance}$: 0.06, 95% CI [−0.08 to 0.19], *P* += 0.82, BF$_{10}$ = 0.05, ESS$_{bulk}$ = 4642.4, Ratio of Sig. Electrodes$_{P+ > 0.95}$ = 31/143; Conflict-Free Estimate$_{approach>return}$: −0.06, 95% CI [−0.21 to 0.11], *P* += 0.24, BF$_{10}$ = 0.05, ESS$_{bulk}$ = 4166.8, Ratio of Sig. Electrodes$_{P+ > 0.95}$ = 11/143; see Supplementary Fig. 3 for an analysis of the brief increase in low frequency power from an event-related potential at the beginning of the approach period in the MFG contacts. We next tested for the existence of theta oscillations in our MFG contacts to assess if the absence of theta power changes between approach and avoidance windows was due to a lack of theta oscillations in this region, or due to inconsistency across

**Table 1 | Analysis Guide. Gives the model specification, number of observations, levels, and exclusion criteria for each model discussed in the Results, along with the corresponding figures and tables**

| Finding | Dependent variable | Fixed effect | Random effect | Number of models | Number of Obs. | Number of levels | Exclusion criteria | Figures/ Tables |
|---|---|---|---|---|---|---|---|---|
| Behavioral result: If the last dot (reward) was large, participants were closer to the ghost at turnaround | distance between Pac-Man and the Ghost at the turn-around point | 2-level factor, specifying if the last dot was large or small | subject | 1 | 34298 | 211 (191 online participants, 20 iEEG patients) | trials where no turn was made; conflict-free trials | Fig. 1d; Source Data |
| Theta power is higher during approach compared to avoidance during conflict trials | Mean theta power during the 1500 ms before/after turn | 2-level factor, specifying avoidance (after turn) or approach (before turn) | subject/electrode | 5, one for each region | 13335–30665 | 15–20/73–167 | conflict-free trials; time periods before the Pac-Man initiated movement; trials where 0 dots were collected; attack trials | Fig. 2; Source Data |
| Theta power is not higher during approach compared to reward during conflict-free trials | Mean theta power during the 1500 ms before/after turn | 2-level factor, specifying return (after turn) or approach (before turn) | subject/electrode | 5, one for each region | 4718–11,146 | 15–20/73–167 | conflict trials; time periods before the Pac-Man initiated movement; trials where 0 dots were collected | Supplementary Figure 2 Source Data |
| Prefrontal and subcortical regions form subnetworks within a wider theta circuit | Average theta coherence during approach period (1500 ms before choice to avoid) for a given region, logged | 4-level factor, specifying partner region | subject/first electrode of pair | 5, one for each region | 1923–4097 | 15–20/115–355 | conflict-free trials; trials where 0 dots were collected | Fig. 3c; Supplementary Figure 5 6 Supplementary Tables 3, 4; Source Data |
| Theta coherence rises (falls) during the approach (avoidance) period | Theta coherence during the approach or avoidance period, logged and scaled | Time | subject/electrode pair | 2, one for approach and one for avoidance period | 59,136 | 20/2816 | conflict-free trials; electrode pairs that did not show elevated theta coherence; trials where 0 dots were collected | Fig. 3d–f; Supplementary Table 5; Supplementary Figure 6 Source Data |
| A subset of region pairs drive the increase (decrease) in coherence during approach(avoidance) | Theta coherence during the approach or avoidance period, logged and scaled | Time*Region Pair | subject/electrode pair | 2, one for approach and one for avoidance period | 59,136 | 20/2816 | conflict-free trials; electrode pairs that did not show elevated theta coherence; trials where 0 dots were collected | Fig. 3g; Source Data |
| The amygdala and MFG drive theta oscillations in the OFC/ACC (Granger) | Average GC during approach period(1500 ms), scaled, not mean centered | Intercept | subject/first electrode of pair | 10, one for each pair of regions | 29–313 | 7–17/20–81 | conflict-free trials; electrode pairs that did not show elevated theta coherence; electrode pairs that did not show 500 ms of elevated directionality; excluded trials where 0 dots were collected | Fig. 4a, b, Supplementary Tables 7 and 8; Source Data |
| The amygdala and MFG drive theta oscillations in the OFC/ACC (cross-correlation) | Time lag with the highest correlation from a cross-correlation analysis between two electrodes, scaled and mean centered | Intercept | subject/first electrode of pair | 10, one for each pair of regions | 50–464 | 10–17/20–112 | conflict-free trials; electrode pairs that did not show elevated theta coherence; time periods before the Pac-Man initiated movement; electrode pairs whose largest correlation was not greater than 95% of a null distribution; trials where 0 dots were collected | Fig. 4a, b; Supplementary Tables 7 and 8; Source Data |
| Theta and HFA synchrony correlate with approach times | Trial-level estimate of synchrony, scaled | Approach time | subject/electrode | 2, one for theta and HFA synchrony | 295,367 | 20/2822 | conflict-free trials, time periods before the Pac-Man initiated movement; trials where 0 dots were collected | Fig. 4c–e; Supplementary Tables 6 and 9; Supplementary Fig. 8; Source Data |
| HFA in the right MFG drops faster during Chase compared to Strike trials | HFA in the right MFG | time*2-level attack factor (Chase or Strike) | subject/electrode | 1 | 123,740 | 9/54 | attack-free trials; time before attack; lMFG electrodes; trials where 0 dots were collected | Fig. 5b–e; Source Data |
| Limbic Theta Power still drops at turnaround, even in Ghost Attack Trials | Mean theta power during the 1500 ms before/after turning | 2-level factor, specifying avoidance or approach | subject/electrode | 5, one for each region | 5404–10,600 | 15–20/73–143 | attack-free trials; time before Pac-Man initiated movement; trials where 0 dots were collected | Fig. 5f, g; Source Data |

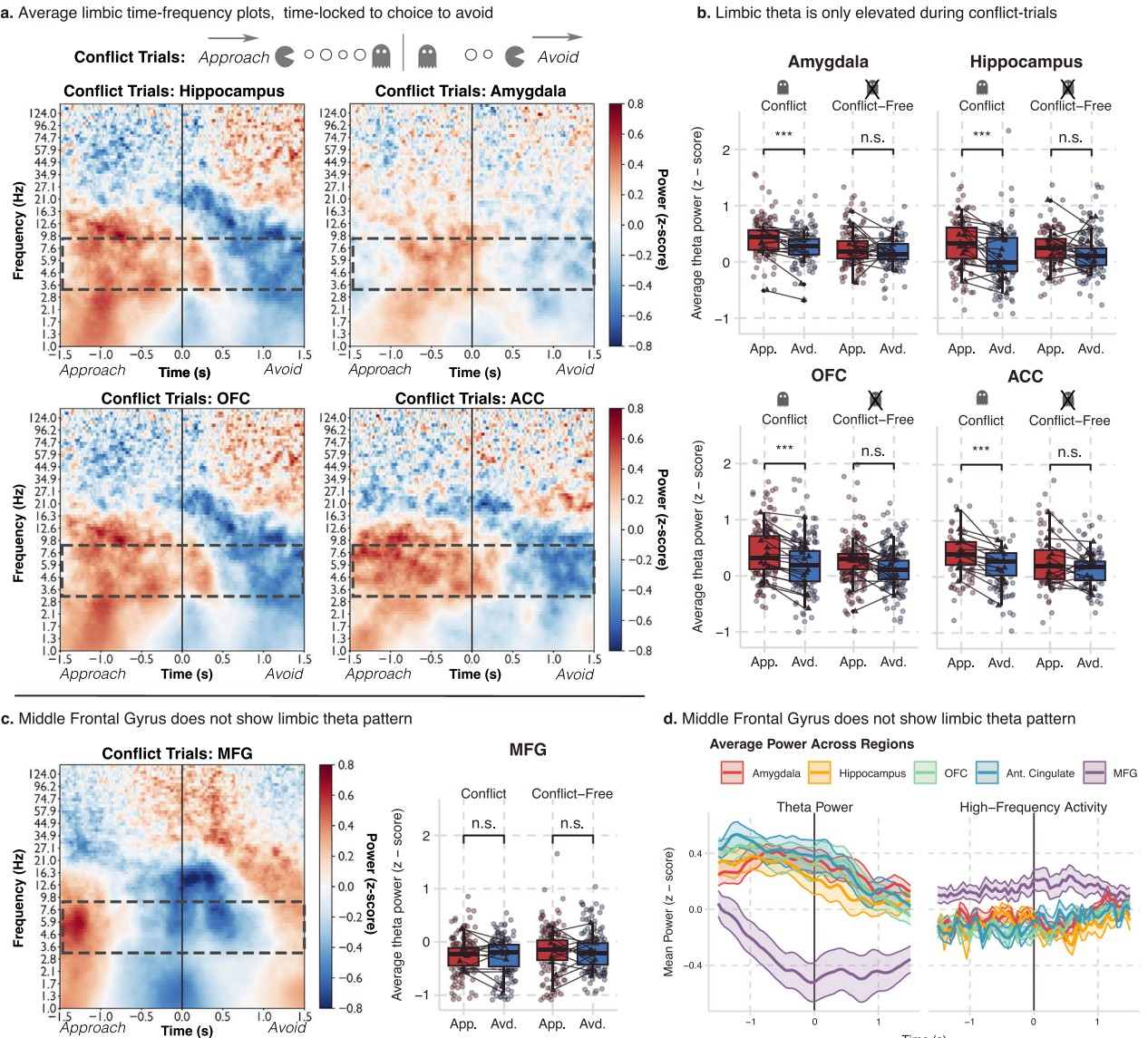

**Fig. 2 | Theta power is elevated during approach in limbic regions, but not MFG. a** Time-frequency analysis of neural activity in conflict trials time-locked to the final choice to stop approaching and begin avoiding across all electrodes in the limbic regions (hippocampus, amygdala, OFC and ACC). Red (blue) indicates increases (decreases) in power. Power was log-transformed and z-scored to the mean power across the 3-second time window. Dotted box indicates the theta band (3-8 Hz) See Supplementary Fig. 2 for time-frequency analysis of the conflict-free trials. **b** Average theta power in the 1500 ms before (called "Approach") and after the choice to turn around on each trial (called "Avoid" or "Return") for each limbic region in conflict-trials (ghost present) or conflict-free trials (ghost absent). Box plots depict median and interquartile range across electrodes ($n_{hc} = 115$, $n_{amyg.} = 101$, $n_{OFC} = 167$, $n_{acc} = 73$), with whiskers covering most extreme values except outliers. Dots

indicate average theta power in electrodes across trials, triangles indicate average theta power across electrodes within a participant. Significance was assessed using a Bayesian linear mixed-effects model with random effects of participant and electrode. Stars indicate that the 95% credible interval did not include 0. **c** Time-frequency analysis of the conflict trials and theta-power analysis for the MFG. Box plots depict median and interquartile range across electrodes ($n_{mfg} = 143$), with whiskers covering most extreme values except outliers. **d** Time-course of theta power (left subplot) and high-frequency activity (HFA, right-subplot) across approach and avoidance windows. Power data was extracted from the TFR and z-scored using the average power in the ITI. Color indicates region, and shading is the standard error of the mean power across patients.

patients or power stability across the trial. We used the FOOOF (Fitting Oscillations and One-Over-F) algorithm—which decomposes the power spectrum into its aperiodic (1/f) and oscillatory components—to these contacts and found evidence of theta oscillations in the MFG in each patient in at least one contact[37]. Therefore, the MFG did have evidence of theta oscillations during our task. However, the dynamics of the theta oscillations did not differ between approach and avoidance periods and were inconsistent across contacts and patients (see Supplementary Fig. 4). Additionally, the MFG was characterized by an elevation of high-frequency activity (70–150 Hz) around the time of choice, not apparent in other regions (See Fig. 2c, d).

## Limbic regions are connected via theta band activity during approach-avoidance decision-making

We hypothesized that the modulations of local theta power were accompanied by increased connectivity between these regions. We calculated the imaginary part of coherence, which removes the interactions due to potentially spurious volume conduction effects[38], between theta activity in pairs of electrodes from different regions. We compared the true imaginary coherence to a shuffled distribution where trial labels in one region were shuffled 1000 times. Each time point was compared to the shuffled distribution and then FDR-corrected for the number of time points. Pairs of electrodes were marked as having

**a.** Theta coherence was compared to a trial-shuffled null distribution

**b.** All three connectivity metrics show similar levels of theta coherence

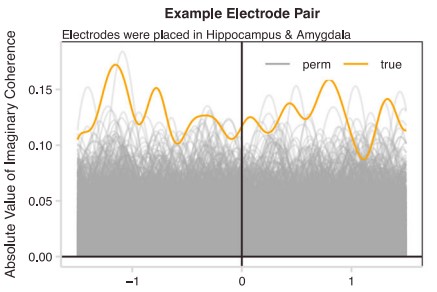
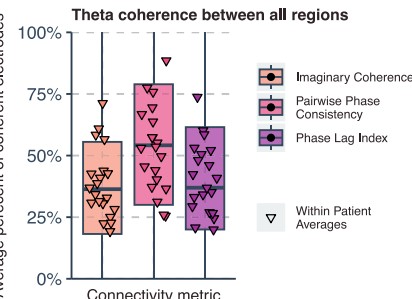

**c.** Prefrontal and subcortical regions form subnetworks within a wider theta circuit

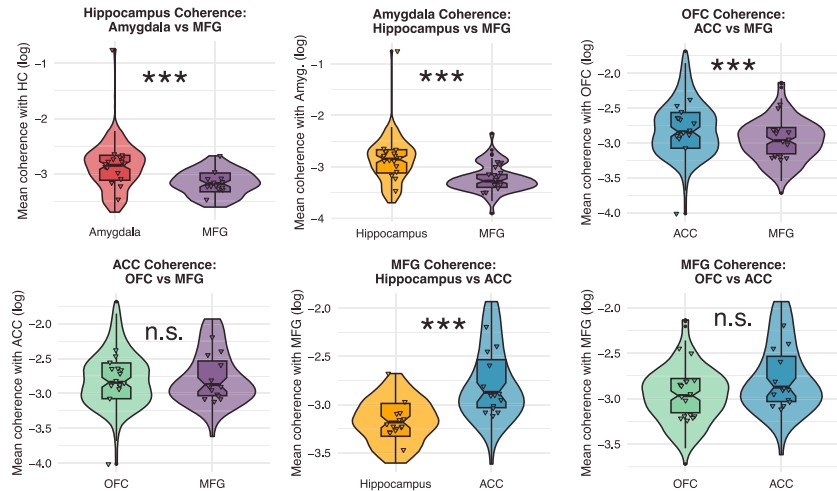

**d-g** PFC-limbic theta coherence rises during approach, falls during avoidance

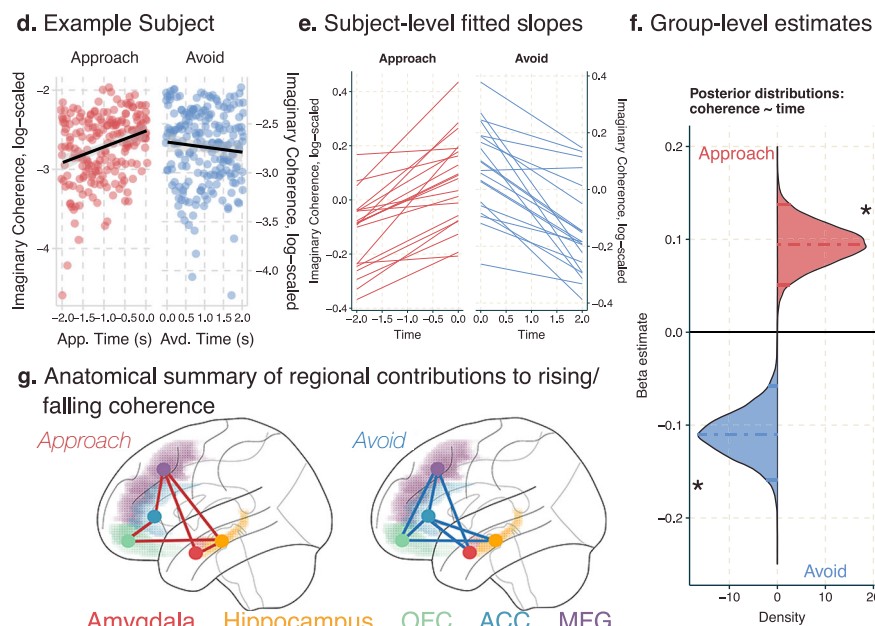

significant theta-band coherence if there was a time window of at least 100 ms of significantly elevated coherence compared to the null distribution (See Fig. 3a and supplement for analyses of additional results across different thresholds). We also calculated two alternative coherence measures, pairwise phase consistency and the debiased estimator of the weighted phase lag index that confirmed our results were robust across different estimates of coherence.

We found more electrode pairs demonstrated coherence within the theta band than would be predicted by chance across each of the limbic regions and with the middle frontal gyrus (See Fig. 3b). We calculated the percentage of electrode pairs that showed elevated coherence in the theta band and averaged this percentage across our 20 patients. In all regions, on average, at least one-third of the pairs

**Fig. 3 | Prefrontal-limbic regions theta coherence during approach. a** Example coherence in an electrode pair from a participant with contacts in the hippocampus and the amygdala. Yellow line indicates the true absolute value of imaginary coherence across the approach and avoidance periods. Grey lines indicate the absolute value of imaginary coherence from the permuted sample, where trial labels for the amygdala contact were shuffled 1000 times. **b** Average percentage of electrode pairs with elevated theta coherence out of all possible pairs for the three connectivity metrics. Box plots depict median and interquartile range across electrodes-pairs ($n_{electrode-pairs}$ = 7302), with whiskers covering most extreme values except outliers. Triangles denote patient-level averages ($n_{patient}$ = 20), color denotes connectivity metric. **c** Violin and boxplots for comparing theta coherence strength between two partner regions within each region of interest. Each facet is a different region of interest ($n_{hc-pairs}$ = 2955, $n_{amyg.-pairs}$ = 2237, $n_{OFC-pairs}$ = 4172, $n_{acc}$ = 2008, $n_{MFG}$ = 3232), and we have highlighted two partner regions to compare on the x-axis (see Supplementary Fig. 4 for all region comparisons). Box plots depict median and interquartile range across electrodes-pairs, with whiskers

covering most extreme values except outliers. Triangles indicate patient averages, *** indicates the 95% confidence interval of the estimated difference in connectivity strength between the two regions did not include 0. **d–g** Increase (decrease) in theta coherence during the approach (avoidance) window. **d** Example patient. Y-axis is the log of the absolute value of imaginary coherence averaged across contacts and region pairs in the approach (red) avoidance (blue) period. X-axis is time in seconds, and each period is fit with a regression line. Shading indicates 95% confidence intervals. **e** Patient-level fitted slopes ($n$ = 20) representing the effect of time on theta coherence in the approach period (red, left) or avoidance period (blue, right). **f** Posterior distributions for the effect of time on theta coherence in the approach period (red) or avoidance period (blue). * indicates the 95% confidence interval of the effect of time on coherence did not include 0. **g** Regional differences in the effect of time on theta coherence in the approach (left) or avoidance (right) windows. Red (blue) lines indicate that the connected regions significantly drove the effect in approach (avoidance) time windows. Colored shading indicates region label.

showed elevated coherence (See Supplementary Table 2), which is higher than the percentage expected by chance. This was comparable across all three measures of connectivity.

We then tested if the consistent limbic theta *power* elevations were indicative of elevated theta *coherence* between limbic regions, as compared to the MFG. To test this, we ran a Bayesian mixed effects model within each region to predict the average theta coherence during the approach window (1500 ms preceding the avoidance decision), using a four-level factor representing the remaining regions as the predictor. This analysis can be thought of as a frequentist ANOVA; however, because the entire model is estimated at once, pairwise differences in coherence strength can be assessed directly without the need for post-hoc tests. We included patients and electrodes as random effects (see "Methods" and Table 1 for details). To avoid inflating our estimates, we tested these models across all electrode pairs, including those that did not show elevated theta coherence in our previous analysis.

Evidence for elevated limbic-to-limbic connectivity compared to limbic-to-MFG connectivity: As predicted, we found that the amygdala exhibited higher theta coherence with the hippocampus compared to the MFG, with an estimated connectivity strength ratio of 1.49 (95% credible interval [1.29, 1.75], $P+ > 0.99$, $ESS_{bulk}$ = 7582.3; See Fig. 3c). This indicates that amygdala-hippocampus coherence was ~1.5x stronger than amygdala-MFG coherence during the approach. Similarly, we found that amygdala-OFC coherence and amygdala-ACC coherence were both stronger than amygdala-MFG coherence, with a connectivity strength ratio of 1.22 (95% CI [1.07, 1.36], $P+ > 0.99$, $ESS_{bulk}$ = 6426.1; see Supplementary Figs. 5) and 1.14 (95% CI [1.03, 1.28], $P+ > 0.99$, $ESS_{bulk}$ = 11399.5; see Supplementary Fig. 5), respectively. We found similar results in the hippocampus (See Fig. 3c; Supplementary Fig. 5 and Supplementary Table 3); hippocampus coherence with the amygdala was 1.41 (95% CI [1.23, 1.65], $P+ > 0.99$, $ESS_{bulk}$ = 5789.6) times greater than its theta coherence with the MFG, and 1.14 (95% CI [1.03, 1.27], $P+ > 0.99$, $ESS_{bulk}$ = 5719.2) times greater with the OFC compared to the MFG. Finally, we found that within the OFC, there is stronger OFC-ACC connectivity compared to OFC-MFG connectivity, with a connectivity strength ratio of 1.2 (95% CI [1.07–1.36], $P+ > 0.99$, $ESS_{bulk}$ = 5742.9). These results provide evidence that the amygdala and hippocampus are more strongly connected to other limbic regions via theta compared to the MFG (See Supplementary Fig. 5 for all regional comparisons using Imaginary Coherence; see Supplementary Fig. 6 for results using Pairwise Phase Consistency and Phase Lag Index; see Supplementary Table 3 for full model results across the three connectivity metrics; see Supplementary Table 4 for posterior predictive checks across connectivity models; see Source Data File for model estimates).

Evidence against elevated limbic-to-limbic connectivity compared to limbic-to-MFG connectivity: Within the ACC, we see stronger theta

coherence with the MFG compared to the amygdala and hippocampus, where ACC-amygdala coherence was only 0.76 (95% CI [0.65, 0.88], $P+ < 0.01$, $ESS_{bulk}$ = 4476.4) times the ACC-MFG coherence and ACC-hippocampus was only 0.77 (95% CI [0.65, 0.90], $P+ < 0.01$, $ESS_{bulk}$ = 3670.9) times ACC-MFG coherence (See Fig. 3c, Supplementary Fig. 5 and Supplementary Table 3 for details). Similarly, the ACC showed preferential connectivity with the OFC compared to the subcortical limbic regions (See Fig. 3c, Supplementary Fig. 5 and Supplementary Table 3 for details), meaning that, while the ACC shared similar theta power modulations as the subcortical limbic regions, the ACC was preferentially connected to other prefrontal regions compared to the two subcortical limbic regions. Furthermore, while the OFC had preferential connectivity with the ACC compared to the MFG, there was no difference in connectivity strength between OFC-hippocampus and OFC-MFG connectivity, nor any difference between OFC-amygdala and OFC-MFG connectivity (See Fig. 3c, Supplementary Fig. 5 and 6 and Supplementary Table 3–4 for details). Thus, instead of finding evidence for increased limbic-to-limbic connectivity, we found that the OFC had a more distributed connectivity pattern, while the ACC was more strongly connected to the OFC and MFG. Contrary to our initial hypothesis, the prefrontal regions, including the MFG, shared elevated theta coherence, indicating the connectivity was not limited to the limbic regions.

Connectivity within the MFG: We examined the connectivity patterns within the MFG. We found that the MFG also had preferential connectivity with the other prefrontal regions (OFC, ACC) compared to subcortical regions (See Fig. 3c, Supplementary Fig. 5 and Supplementary Table 3 for details), matching the profile we found in the ACC.

## Theta coherence increases over the approach period and falls during avoidance

Next, we examined dynamic task-related changes in this circuit. Rodent models report that theta synchrony between the hippocampus and mPFC increases in anxiogenic versus safe contexts[25]. We hypothesized that theta synchrony across the prefrontal-limbic network would increase across the approach period as patients increased both the level of reward, threat, and conflict.

We tested this by using time to predict theta coherence in the two seconds preceding the choice to avoid (see Methods and Table 1 for details). We averaged the connectivity values by region pair (e.g., OFC-ACC, Amyg-Hipp., etc.) within patients and within 100 ms time bins. We again used Bayesian mixed-effects models with a random effect of patient and electrode pair and limited electrode pairs to only those that showed significant levels of theta coherence in either the approach or avoidance periods to ensure we were capturing fluctuations in true coherence as opposed to random noise. We found that average connectivity values increased during the time the patient approaches (Estimate: 0.09, 95% CI [0.05–0.14], $P+ > 0.99$, $BF_{10} > 9$,

$ESS_{bulk} = 7680.4$, Ratio of Sig. Electrodes$_{P+ > 0.95} = 1782/2816$; See Fig. 3d-f, Source Data File). While local theta power increases might potentially inflate connectivity estimates, this is unlikely the case because theta power—even though initially elevated—drops or remains constant at the end of the approach period, corresponding to when coherence across the network is highest (See Fig. 2d). We confirmed this result using two other connectivity metrics (Pairwise Phase Consistency, Phase Lag Index) and across four different time windows ranging from 1.5-2 seconds (Supplementary Table 5). We next included an interaction term between time and region pairs to identify which region pairs were driving this effect. We found that the following pairs demonstrated strong time-coherence modulations: amygdala-MFG, amygdala-hippocampus, hippocampus-MFG, hippocampus-OFC, ACC-OFC, and ACC-MFG (See Fig. 3g; Source Data File). Next, we tested if theta coherence would fall, remain constant, or continue to rise during the avoidance period. We hypothesized that the coherence would fall when retreating from the ghost, indicating that theta coherence across the circuit is highest during the portion of the trial with the highest levels of risk and conflict. As hypothesized, we found that theta coherence drops after the patients began avoiding and moved to exit the trial (Estimate: $-0.11$, 95% CI [$-0.16$ to $-0.06$], $P + < 0.01$, $BF_{10} > 14$, $ESS_{bulk} = 3520.3$, Ratio of Sig. Electrodes$_{P+ > 0.95} = 1824/2816$; see Fig. d-f; see Supplementary Fig. 5; Source Data File), and this effect was driven by the following pairs: amygdala-MFG, amygdala-ACC, hippocampus-ACC, hippocampus-OFC, ACC-OFC, ACC-MFG, and OFC-MFG; See Fig. 3g; Source Data File). As before, we found that the drop in theta coherence during avoidance was consistent across both alternate connectivity measures and multiple timepoints (See Supplementary Table 5).

### Amygdala and lateral prefrontal activity drive theta oscillations in the OFC and ACC

We next assessed the directionality of the prefrontal-limbic circuit using two independent methods, State-Space spectral Granger causality and cross-correlation analysis[25,39]. We calculated the net spectral Granger causality between all electrode pairs with significant levels of theta-band coherence. We implemented the Granger analysis in a pairwise manner to limit the electrodes to those with elevated theta coherence, as well as to better match our control analysis, which used cross-correlation (see Supplementary Tables 7, 8 for estimates of results across different thresholds). As there may be bidirectional information flow, we identified which signals dominated the information flow by calculating the difference between the Granger values calculated in both directions. We next subtracted the spectral Granger values calculated from the reversed time-series, as noise can inflate connectivity estimates if the signal-to-noise-ratio SNR increases or decreases over time[40]. This final net Granger score was compared to a permuted distribution where trial labels from one electrode were shuffled 1000 times. By comparing the absolute value of the true net Granger score to the permuted null distribution, we calculated permuted $p$ values and then FDR-corrected these values for the number of timepoints. If more than 50 timepoints (500 ms) were significant at $p < 0.05$, then that electrode pair was included in our subsequent analyses (see "Methods" and Table 1 for details).

When net Granger scores were higher than chance, we tested for a bias in directionality between our regions. We turned to our linear mixed-effects models to account for shared variance within electrodes and patients. Specifically, we modeled the net Granger scores during the approach window using an intercept-only model and random effects of electrode and patient. From these models, we report the Probability Positive (P+), which represents the proportion of posterior samples where the effect is greater than zero, as a measure of directional certainty, and we consider $P + > 0.95$ as strong evidence for directionality. Since we did not have strong a priori hypotheses about the directionality in this circuit, we also validated these analyses by conducting a cross-correlation analysis on the theta-band passed signals for each pair of cohering electrodes (see "Methods" and Table 1 for details). We used similar linear mixed effects models to assess the P+ of the lead and lag times resulting from the cross-correlation analysis. Results were only interpreted if they resulted in significant P+ values using both independent methods, which relied on different calculations of theta power (Morlet Wavelets vs Bandpass filtering) and slightly different time windows (the cross-correlation analysis excluded timepoints where the patient was still at the beginning of the trial, while the Granger analysis included these timepoints). Details of both analyses can be found in the Methods.

While we found evidence of bidirectional information flow in many individual pairs of electrodes, we found four region pairs with consistent directionality across our patients. First, we found that theta oscillations in the Amygdala drove theta in both the OFC ($P_{Granger} + = 0.97$, $ESS_{Granger} = 4891$; $P_{CCF} + = 0.96$, $ESS_{CCF} = 7542$) and ACC ($P_{Granger} + = 0.97$, $ESS_{Granger} = 5788$; $P_{CCF} + = 0.96$, $ESS_{CCF} = 7783$). Second, we found that the Middle Frontal Gyrus also led theta oscillations in the OFC ($P_{Granger} + = 0.99$, $ESS_{Granger} = 5216$; $P_{CCF} + = 0.96$, $ESS_{CCF} = 10048$) and ACC ($P_{Granger} + = 0.99$, $ESS_{Granger} = 7776$, $P_{CCF} + = 0.97$, $ESS_{CCF} = 8239$), meaning that both the OFC and ACC were being driven by amygdal and lateral prefrontal contributions. We did not find consistent evidence of directionality between any of the other region pairs (See Supplementary Table 7 for ratios of significant electrodes, results for other regions, and results for the cross-correlation analysis).

### Pairwise synchrony during approach behavior correlates with choice to avoid

We hypothesized that theta synchrony across the prefrontal-limbic circuit during approach would correlate with the time spent approaching, given rodent data suggesting a role for theta in approach-avoidance[24,25]. Coherence measures were not reliable at the trial level, and we instead calculated the Pearson correlation between the amplitude envelopes of the theta signals in each electrode pair during the approach period, which is a common method for assessing functional connectivity between regions[41]. We tested this using Bayesian mixed effects models with random effects of patient and electrode in the set of electrode pairs that showed significantly high theta coherence. Times were log-transformed and scaled, and synchrony scaled before estimating the model (see "Methods" and Table 1 for details). As hypothesized, the model found an overall significant correlation between approach times and theta synchrony (Estimate: 0.07, 95% CI [0.04–0.09], $P + > 0.99$, $BF_{10} > 100$, $ESS_{bulk} = 2816.6$), with many individual region pairs contributing to this effect (See Fig. 4 c-e; see Supplementary Table 6 for full results; see Supplementary Fig. 8 for results across different theta coherence thresholds; see Supplementary Table 9 for posterior predictive checks; see Source Data File for model estimates). In particular, theta synchrony between the OFC and all other region pairs correlated with longer approach times.

These data provide evidence that a distributed prefrontal-limbic theta circuit is recruited during approach-avoidance conflict, with increased theta coherence associated with longer approach times. We hypothesized that theta power was providing a mechanism for network communication and next tested if local neuronal firing[42,43], as approximated by high-frequency activity (HFA), also correlated with approach times. We used the same high theta coherence electrodes and calculated the Pearson correlation of the high-frequency activity during approach between the electrodes. Times were also log-transformed and scaled, and synchrony was scaled before estimating the model (see "Methods" and Table 1 for details). While the model did not find an *overall* significant correlation between approach times and HFA synchrony (Estimate: $-0.01$, 95% CI [$-0.04$ to 0.01], $P + = 0.1$, $BF_{10} = 0.01$, $ESS_{bulk} = 2159.4$), we did find a more restricted set of region pairs that correlated with approach time (See Fig. 4c–e see

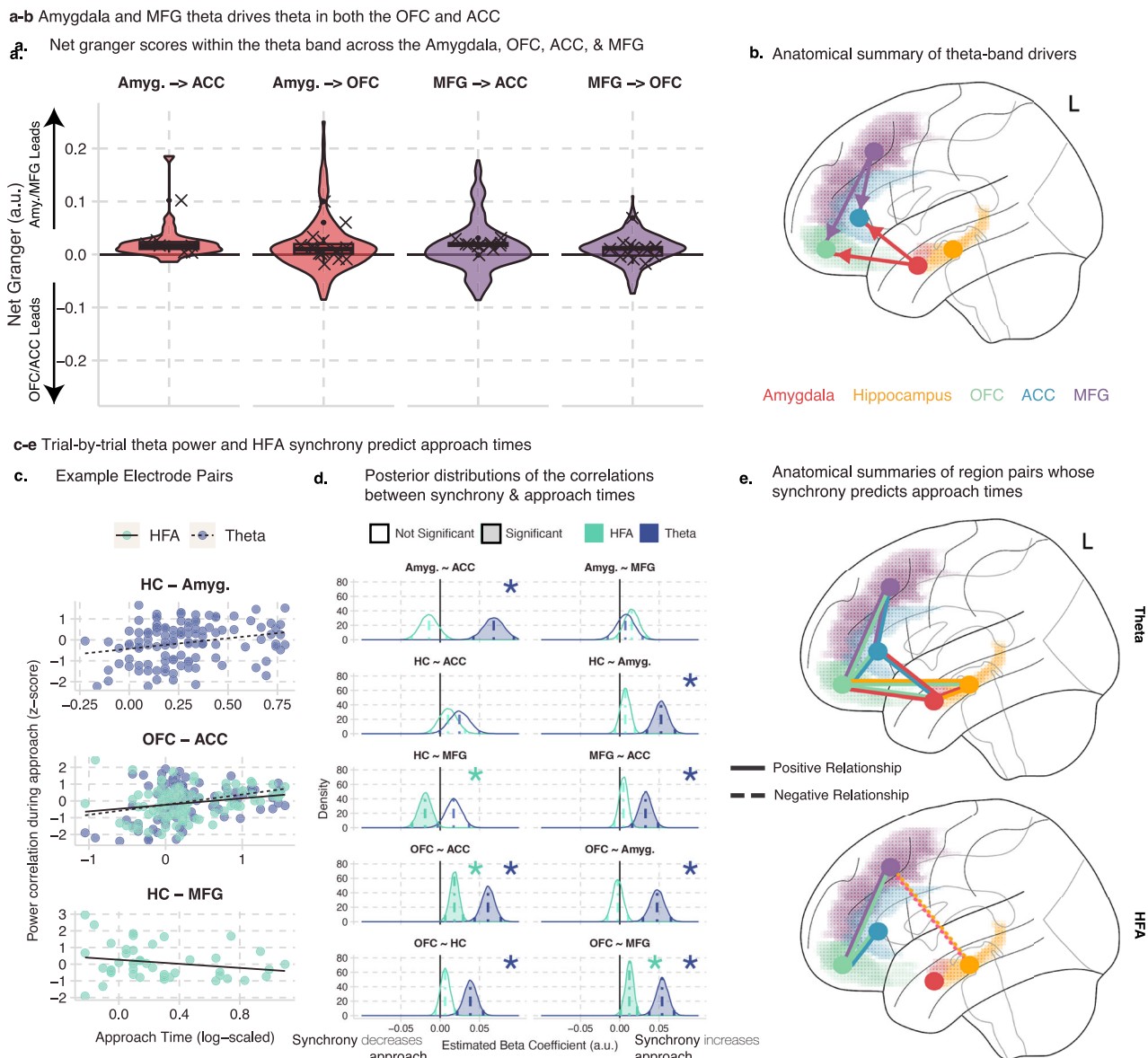

**Fig. 4 | Drivers within the prefrontal-limbic theta circuit and its association with approach duration. a**, **b** Results from the spectral Granger causality (GC) analysis between 4 sets of regions. **a** Y-axis shows net GC, and each facet shows a region pair. Violin and boxplots show the average granger values across the approach window. Box plots depict median and interquartile range across electrodes-pairs, with whiskers covering most extreme values except outliers. Crosses show the patient averages (*n* = 20). Positive values indicate that either the Amygdala or MFG led, while negative values indicate that either the OFC or ACC was the leading region. **b** Anatomical summary of the directional connectivity analysis. Color corresponds to region, and arrows denote the directionality. **c**–**e** Trial-by-trial theta and HFA synchrony correlates with approach times. **c** Association between approach times and theta/HFA synchrony in example pairs of electrodes. X-axis is the untransformed time that a participant spent approaching on a given trial. Y-axis

is the z-scored correlation of theta power during the approach period between the two contacts. Each facet is from a different example pair, and color indicates frequency band. **d** Posterior distributions of the estimated correlation between approach times and either theta power (blue) or HFA (green) synchrony. Linear mixed-effects models were fit using MCMC sampling to estimate the association between approach times and theta synchrony. Colored, dotted lines indicate the mean of the posterior distribution, and solid color lines indicate the 95% credible intervals. Both stars and transparency indicate if the 95% credible intervals include 0. **e** Anatomical summary of the results presented in (**d**). Lines indicate that the connected regions correlated with approach times. Solid lines indicate a positive relationship with approach time; dotted lines indicate a negative relationship. Top (bottom) brain shows results for theta-band (HFA) results.

Supplementary Table 6 for full results; see Supplementary Fig. 8 for results across different theta coherence thresholds; see Supplementary Table 9 for posterior predictive checks; see Source Data File for model estimates). Specifically, we found that OFC-MFG and OFC-ACC high-frequency synchrony correlated with longer approach times. We also found a negative correlation between hippocampus-MFG high-frequency synchrony and approach times, indicating that the more similar the high-frequency activity in the hippocampus and MFG, the more likely the patient was to turn and avoid attack earlier.

## Network circuit reorganization after decision to avoid

We provide evidence of a theta-mediated circuit across cortico-limbic regions, which increases in connectivity strength as reward, threat, and conflict increase. Additionally, we have found that synchrony in both the theta band and HFA in these regions correlates with approach times and that the OFC connects the prefrontal regions with the hippocampus and amygdala. Next, we aimed to understand how this system changes as the threat becomes more proximal. It has been proposed that separate neural circuits and corresponding behaviors

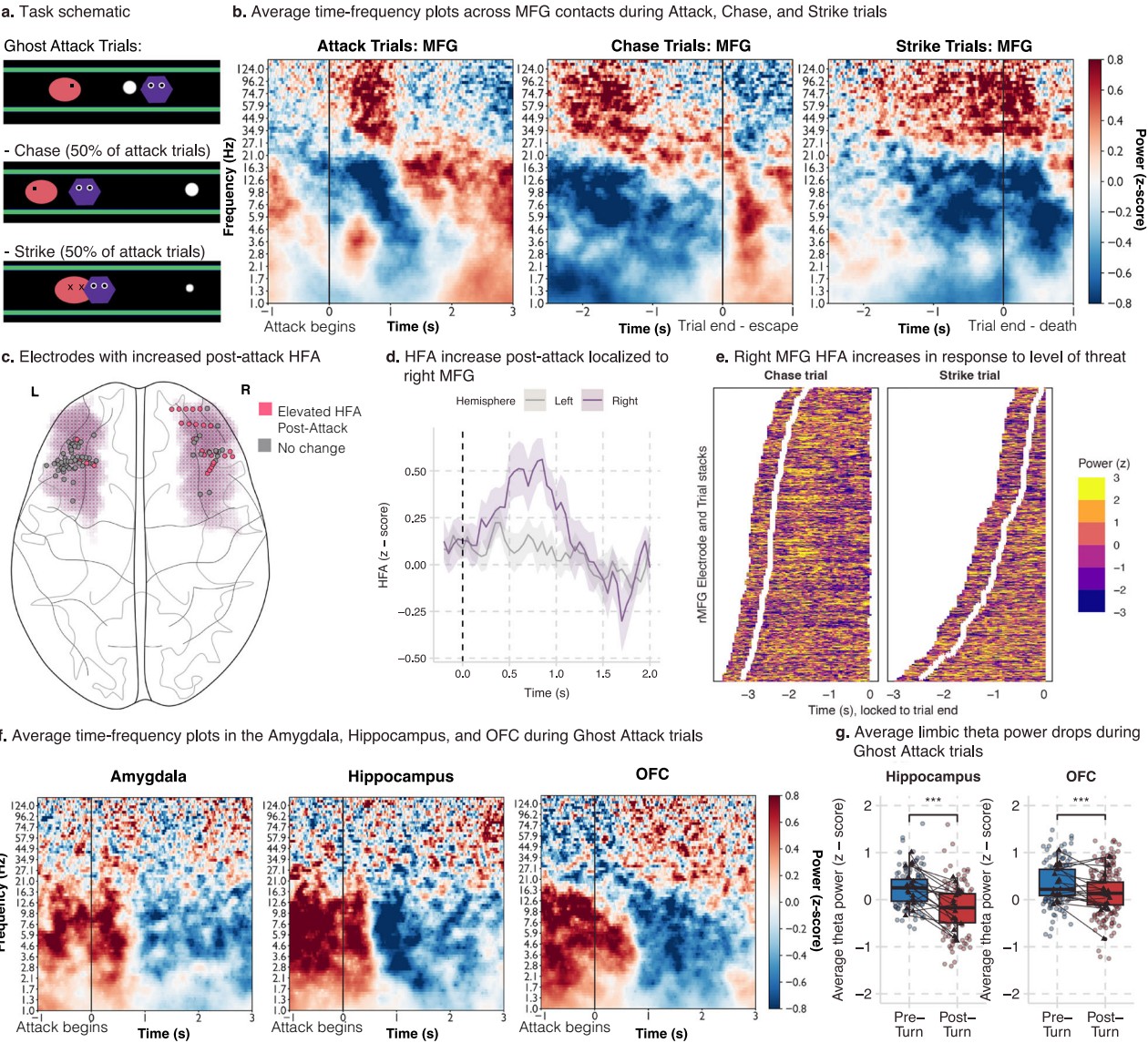

**Fig. 5 | Right Middle Frontal Gyrus HFA increase in response to proximal threat.**
**a** Task schematic of Ghost Attack. On each trial, if the participant moved within a certain distance to the ghost (purple hexagon), they triggered a ghost attack. During Attack, the ghost moves toward the Pac-Man (pink oval). On Attack trials, there is a 50% chance of a Chase or a Strike trial. On Chase trials, the ghost's speed does not change, allowing the participant to respond and safely exit the trial. On Strike trials, the ghost's speed is increased such that they will be caught by the ghost. In the task, participants viewed alternate graphics inspired by the arcade game Pac-Man. **b** Time-frequency analyses of Ghost Attack, Chase, and Strike trials average across MFG contacts and participants. Red (blue) indicates increases (decreases) in z-scored power. The left plot is time-locked to the onset of a ghost attack and includes all attack trials. The middle plot is time-locked to trial end and includes only Chase trials. The right-most plot is time-locked to trial end and includes only Strike trials. **c** Anatomical localization of the increase in HFA after the onset of a ghost attack. Pink shading indicates the MFG, dots indicate contacts, color indicates if a given contact had significantly elevated HFA post-attack.

**d** Average HFA during a ghost attack in the left (grey) and right (purple) MFG contacts. *Y*-axis is z-scored power, and the *X*-axis is time, locked to the onset of the attack. Shading indicates the standard error of the mean across participants. **e** Trial and electrode stacks of right MFG contacts during Chase trials (left plot) and Strike trials (right plot). Brighter (darker) colors indicate higher (lower) power. White dots indicate the onset of the ghost attack, and the *X*-axis is time-locked to trial end. **f** Time-frequency analysis of Ghost Attack trials. Red (blue) indicates increases (decreases) in z-scored power. The *X*-axis is time-locked to the onset of the ghost attack. **g** Average theta power in the 1500 ms before and after turning in response to onset of ghost-attack. Box plots depict median and interquartile range across electrodes ($n_{hc}$ = 115, $n_{OFC}$ = 167), with whiskers covering most extreme values except outliers. Dots indicate average theta power in electrode across trials, triangles indicate average theta power across electrodes within a participant. Significance was assessed using Bayesian mixed effects models, and *** indicates the credible interval did not include 0.

respond to distal threats (anxiety-like behaviors) compared to proximal threats (fear-like behaviors)[44,45]. We hypothesized that we would see the prefrontal-limbic circuit adapt as threat increases. We tested this by investigating the special case of avoidance induced by a Ghost Attack (see Fig. 5a for a trial schematic of these events). During an attack, there were two possible outcomes. First, the ghost could initiate a "Chase", where the ghost's speed remained constant in the

direction of Pac-Man. In this case, as long as the patient turned to avoid, they had a chance to successfully escape. Second, the ghost could initiate a "Strike", where the ghost's speed was increased such that even if the patient responded appropriately and fled the attack, they would be caught by the ghost and lose a life. "Chase" versus "Strike" trials were counterbalanced, while the advent of an attack was determined by the patient's behavior in the trial. All attack trials were

removed before assessing regional theta power modulations to ensure that the effects described in Fig. 2 were not driven by this subset of trials.

We time-locked to ghost attack initiation and found similar drops in theta power in the OFC, anterior cingulate, hippocampus, and amygdala as observed during distal-threat avoidance, despite these trials being excluded in those earlier analyses (See Fig. 5f, g, Source Data File). We next turned to the MFG, where calculating the average TFR showed a striking increase in the HFA after the ghost initiates an attack (see Fig. 5b). However, a Bayesian mixed effects model predicting HFA in the 1 second after the attack compared to the second before the attack found that this increase was driven by only a subset of electrodes (39/127 total MFG contacts). Given that the effect within this population of electrodes was strong enough to dominate our TFRs (See Fig. 5b), we assessed if these electrodes were anatomically distinct. We found that the majority of the attack-activated electrodes were localized to the right hemisphere (84%; See Fig. 5c). When we included hemisphere as an interaction in our HFA model before and after ghost attack, we found a significant interaction between hemisphere and attack, where right MFG contacts resulted in a significant elevation of HFA in the second after an attack began (estimate: −0.16 95% CI [−0.29 to −0.04], P + <0.01, $BF_{10}$ > 1.75, $ESS_{bulk}$ = 6782.9; See Fig. 5d; Source Data File). To characterize what was driving HFA during the attack, we limited our analyses to the contacts in the right MFG (See Table 1 for model details). We hypothesized that HFA would correlate with the level of threat, where activation would decrease if the patient realized they could successfully escape. To test this hypothesis, we ran a Bayesian mixed effects model where time was used to predict HFA on the trial level (See Table 1 for model details). We included an interaction term for trial type (chase or strike trials), where we predicted chase trials to have a negative relationship between time and HFA as patients realized they would make a successful escape. As predicted, we found a main effect of trial type, where chase trials decreased HFA compared to strike trials (Estimate: −0.16 95% CI [−0.21 to 0.12]], P + <0.01, $BF_{10}$ > 100, $ESS_{bulk}$ = 13160.1; see Source Data File) and a significant interaction with time where chase trials showed dropping HFA as patients neared the end of the trial (estimate: 0.12 95% CI [−0.17 to −0.08], P+ <0.01, $BF_{10}$ > 100, $ESS_{bulk}$ = 13324.6; See Fig. 5e; see Source Data File). We interpret these results as evidence that high-frequency activity in the right MFG tracks the perceived threat of the ghost. Given that ghost attacks were designed to motivate the behavior on attack-free trials, we could not investigate the connectivity between the MFG during these trials, as there were only an average of 58+/−13.4 trials (across both chase and strike trials) in the patients.

## Discussion

We provide behavioral and electrophysiological evidence for a dynamic cortico-limbic network supporting approach and avoidance. We identified approach and avoidance-dependent modulations in theta power, where theta power was elevated during the approach window and dropped during avoidance in the hippocampus, amygdala, OFC, and ACC. In contrast, in the middle frontal gyrus there was no evidence of theta-band power modulations during approach, but there was increased high-frequency activity around the time of turnaround from the Ghost attack. We also found increased connectivity in the prefrontal-limbic circuit, with increased theta-band coherence across the four limbic regions and the MFG. Network theta coherence increased during the approach period, as reward, threat, and conflict increased, but fell as patients turned to avoid the ghost. We also observed that theta oscillations in the OFC and ACC were driven by MFG and amygdala activity, suggesting OFC and ACC integrate information from both regions. The degree of synchrony between the nodes in the circuit correlated with trial-dependent changes in approach behavior, where higher theta synchrony between multiple

regions, but especially with the OFC, correlated with longer approach times. In contrast, increased high-frequency synchrony between the hippocampus and MFG correlated with shorter approach times. Finally, during attack trials, when the threat on a given trial shifted from distal to proximal, high-frequency activity in the right MFG tracked threat level. Together, these findings support engagement of multiple components of the prefrontal-limbic network during anxiety-inducing behavior.

Low-frequency theta oscillations coordinate neural activity between distal regions[46–48], facilitating information transfer[49,50]. Absent between-region synchrony, inputs may arrive at random phases of the excitability cycle, hindering effective inter-regional communication. Notably, the degree of theta coherence correlates with behavioral performance in a wide range of attention, learning, emotional, and working-memory tasks[51–58]. Our results support a key role for theta synchrony during approach-avoidance. We found many theta-cohering pairs of electrodes across the prefrontal-subcortical circuit with theta band synchrony peaking at the final decision to avoid– when the levels of threat, conflict, and the need for control were the highest. The prefrontal cortex is proposed to have a role in coordinating the oscillatory activity underlying cognitive control and goal-directed behavior[47], and midline theta oscillations recorded from scalp EEG have been long-associated with uncertainty and cognitive control[59,60]. Furthermore, in the context of approach-avoidance, Amemori et al. reported that reduced connectivity in a PFC-limbic circuit results in more avoidant behavior in non-human primates[19]. In line with these results, we found that the MFG had Granger causal influence on theta oscillations in both the OFC and ACC, and increased theta synchrony across multiple nodes in the prefrontal-limbic theta network correlated with longer approach times in the task, meaning that patients took on more risk to receive increased reward in those trials. Importantly, while Amemori focused on alpha oscillations in a restricted set of limbic regions, they defined the alpha band as between 5 and 13 Hz, and reported peaks at 7 Hz, which overlaps with our 3−8 Hz theta band definition for this study.

However, while there is converging evidence that theta oscillations provide a cross-region mechanism for neural communication and that theta oscillations are prominent during approach-avoidance decision-making, there are open hypotheses about what kind of information is being coordinated across regions via theta oscillations. Approach-avoidance behavior is dependent on a multitude of functions, such as threat detection, uncertainty estimation, and conflict resolution between competing goals[20,61]. For example, in our task, increased theta oscillations and coherence could reflect increased volitional cognitive control to successfully obtain more reward in the face of increasing task demands. Alternatively, the accumulating risk and conflict over the trial could drive increasing theta synchrony in a bottom-up manner. The Ghost attack trials are informative as we observe a drop in theta power after the turn to avoid a direct attack by the Ghost, similar to the decrease in theta power during an attack-free trial. Alternatively, we could have seen theta oscillations continue or even increase as the Ghost attacks and gets closer to the Pac-Man, which would suggest a greater correspondence between risk or anxiety and theta oscillations. Instead, theta power falls approximately one second after the onset of the Ghost Attack, corresponding to after the participant has turned to avoid the attack. This is compatible with a role for theta oscillations in navigating uncertainty, conflict, or exerting cognitive control– before an attack, the ghost needs to be closely monitored, but after an attack and a turn is made, the need to control behavior concludes. Furthermore, our findings that theta coherence drops during the avoidance period support the idea that theta oscillations track uncertainty or cognitive control–while there may still be conflicting goals after turning to avoid due to remaining reward in the trial, theta coherence begins dropping during this time, suggesting a decreased

need to predict the ghost's actions (uncertainty) and to adjust one's behavior in response to those predictions (cognitive control). Computational models are needed to fully test these interpretations, but the current results are in accord with work showing a role for theta oscillations handling uncertainty in the ACC, especially when that uncertainty requires behavioral adaptation[62,63]. Given that the OFC and ACC were the main receivers of theta oscillations in our task, they may be well-suited to integrating information from the other regions in order to respond to uncertainty.

Our HFA synchrony results also provide insight into what information may have been carried via theta oscillations. While the OFC-MFG and OFC-ACC HFA synchrony correlated with longer approach times, hippocampus-MFG HFA synchrony correlated with shorter approach times. This indicates that patients were more likely to forgo reward in favor of lower risk when the MFG and hippocampus had similar HFA profiles. This aligns with previous studies using approach-avoidance paradigms in rodents, which have found the hippocampus to be a key node within this circuit[61,64,65] (for a recent review, see ref. 20). Studies in both rodents and humans have proposed that threat information is propagated from the hippocampus to the prefrontal cortex via theta during approach-avoidance[24,30] Padilla-Coreano et al. found that disrupting hippocampus to mPFC theta synchrony reduced avoidance, but increasing synchrony increased avoidance behavior in mice. Further analyses revealed that it was not the synchrony itself that determined the avoidance behavior, but the degree of information transmission from the hippocampus to the mPFC that biased the animals toward avoidance. Furthermore, recent work suggests that high-frequency power correlations reflect functional, but not direct, connectivity[47,66], meaning that the similar HFA profiles in hippocampus and MFG may result from the coordination by a third region, rather than the result of direct neuronal interaction. Supporting this interpretation, we found lower theta synchronization between the hippocampus and MFG. Instead, we observed that the OFC and ACC share high theta coherence with both the subcortical regions and the MFG, suggesting that these regions may be integrating the information across distinct neural circuits. This has been hypothesized for the ACC under reward and threat processes[62,67,68], where ACC-Amygdala connectivity is reported to track uncertain threat[69], while PFC-ACC connectivity may relate to emotion regulation[18,70]. Our directionality analyses support these reports with both amygdala and lateral prefrontal activity driving theta oscillations in the OFC and ACC. Thus, while similar representations of threat information in the hippocampus and MFG may lead to early avoidance, other types of information, such as reward or uncertainty, may also propagate through the circuit. Accordingly, other region pairs, such as the OFC-ACC and OFC-MFG had positive relationship between HFA synchrony and approach times.

Fitting with a model of approach-avoidance behavior where distal and uncertain threat results in anxiety-like behavior while proximal and certain threat induces fear behaviors[44,45], we see rapid changes in the cortical-subcortical circuit during ghost attack. Specifically, the time-frequency plots were dominated by a rapid increase in high-frequency activity in the right MFG during ghost attack. Previous work has reported evidence of left/right asymmetry in the lateral PFC, including the MFG, especially in the context of emotional regulation[71,72]. It has been shown that the right dorsolateral PFC is activated during periods of unpredictable threat, and this increased dlPFC activation is associated with reduced anxiety[73]. These authors interpret their results as the right dlPFC enacting emotional regulation by providing top-down control on the limbic regions. In our study, the high-frequency activity in the right MFG during the acute threat period may provide a control signal as patients try to regulate their escape in the face of a near-term threat. Once patients realize they will make a successful escape, the need for

this control drops and is accompanied by a rapid decrease in high-frequency activity. Alternatively, or perhaps in addition to a role for emotional regulation, the activity in the MFG in this task may reflect conflict or danger expectation, as has been shown in other studies investigating approach-avoidance and other types of conflict[30,35,74]. MFG-limbic theta coherence increased during the approach period, and dropped during avoidance, particularly with the amygdala, and the MFG Granger causal theta oscillations in the OFC and ACC during the approach period. It is possible that threat probability is calculated or encoded in MFG neurons and then transmitted to the OFC and ACC via theta oscillations to the OFC and ACC, which integrate that information with other signals to guide behavior.

While regional and inter-regional contributions to approach-avoidance behavior have been identified across human, NHP, and rodent studies, this is the first human study characterizing the theta-mediated circuit dynamics enabling these regions to function in concert in real-time. Human fMRI and NHP research have highlighted the crucial role of widespread prefrontal involvement, while rodent studies have implicated mPFC-subcortical theta synchrony in regulating approach and avoidance behaviors. Our findings integrate this literature by demonstrating that theta coherence links both subcortical structures with prefrontal regions, but also links both the cognitive and limbic regions within the prefrontal cortex. The circuit architecture identified here suggests that approach-avoidance behavior is governed by hierarchically organized theta-based interregional networks. For instance, the hippocampus and amygdala exhibit robust theta coherence, but amygdala signals propagate to both the OFC and ACC, allowing integration with input from the MFG. Importantly, by examining this behavior in real time, we show that network strength increases as the individual reaches the decision point between approach and avoidance, suggesting that this circuit is most engaged when threat and conflict intensifies.

There are caveats to this research. First, our study was conducted on patients with epilepsy undergoing clinical evaluation, raising the question of the generalizability of our findings. To address this, we undertook extensive efforts to only test patients fully alert and cooperative and excluded electrodes near seizure foci or contaminated by artifacts. We also show similar behavioral patterns across our patients and an online sample of 191 participants (See Fig. 1, panels b–e). It is important to note that electrodes were placed based solely on the clinical needs of the patient. As such, coverage varied across patients, and some effects may have been missed due to reduced coverage. We attempted to control for this by using hierarchical models that accounted for within-patient and within-region correlations, as is considered best practice[75]. Similarly, coverage limitations prevented us from analyzing subregion effects, such as between the anterior and posterior hippocampus. The rodent literature indicates that the ventral hippocampus, corresponding to the anterior hippocampus in humans, is involved in approach-avoidance, as we observed[20,61]. However, we were not powered to test for anterior versus posterior hippocampal effects.

In conclusion, we used intracranial recordings and a continuous-choice approach-avoidance task in humans to characterize prefrontal-limbic neural activity during anxiety-inducing decision-making. We found broad regional involvement with a particular focus on theta power modulations and coherence supporting a distributed prefrontal-limbic network underlying approach-avoidance. The findings contribute to a circuit understanding of psychiatric disorders, especially those characterized by maladaptive avoidance, such as generalized anxiety disorder, agoraphobia, and social anxiety disorder.

## Methods

### Patients

We recorded intracranial signals from twenty (10 female) adult patients (mean age 27.25+/−12.04) who underwent intraoperative neurosurgical treatment for pharmacoresistant epilepsy. All patients were implanted with depth electrodes to localize the seizure onset zone for eventual surgical resection. We selected patients with implantations in any of the following regions: hippocampus, amygdala, OFC, anterior cingulate, and middle frontal gyrus. The electrode placement was dictated solely by the patient's clinical team. See Supplementary Fig. 1 for patient specific electrode implantation and see Supplementary Fig. 9 for a characterization of their self-reported depression and anxiety symptoms. Sex was self-reported and not considered in the study design due to opportunistic sampling strategy. The recordings took place at the following hospitals: Loma Linda University Medical Center ($n = 6$), Barnes-Jewish Hospital in St. Louis ($n = 12$), and the St. Louis Children's Hospital ($n = 2$). We additionally collected 191 participants from the online recruitment platform, Prolific, using the "Representative sample" option, which balances the sample with regard to demographics like sex, based on census data from the US. The online participants were paid a set amount for their participation in the research, along with a bonus that correlated with task performance. Participants were given thorough instructions along with a short quiz to ensure comprehension of the task. All patients provided written informed consent as part of the research protocol approved by each hospital's Institutional Review Board and by the University of California, Berkeley. Patients were tested when they were fully alert and willing to participate.

### Behavioral task

On each trial, Pac-Man, the Ghost, and dots are placed along a single corridor that runs horizontally across the screen. The participant controls the Pac-Man, and they are instructed to collect as many dots as they can while avoiding the Ghost, which is moving back and forth at a constant speed at the opposite side of the corridor. The participant makes a single button press, using the left or right arrow keys, to initiate movement along the corridor at a constant speed. By moving towards the center of the corridor, they can collect up to 5 dots, resulting in points, which varied in either small (10 points) or large (20 points) sizes and are placed in pseudorandom configurations between Pac-Man and the Ghost. Importantly, at any point in the trial, they could also choose to move away from the center of the corridor (and, thus, away from both the dots and the Ghost), where they could exit to the next trial at the end of the corridor. If Pac-Man was "caught" by the Ghost, they lost one of their three lives, they also lost all the points acquired on that trial, and the Ghost's capture was accompanied by a "death animation" used in the arcade game (e.g., a loud sound and the Pac-Man shape rotates and disappears). The Ghost's starting direction and location, the distance between the dots and Pac-Man, and Pac-Man's starting location are counterbalanced across trials.

The Ghost had three states, which we term "Pace", "Chase", and "Strike". The default state was "Pace", where the Ghost moves in a set path back and forth at the end of the corridor. The end of the path was a jittered distance away from the last (5th) dot. In this state, the Ghost moved at the same speed as the Pac-Man and did not deviate from its path to capture the Pac-Man. The remaining two states, which we jointly call a "Ghost Attack", were designed to motivate behavior on a typical trial (See "Results" for Details). On these Attack trials, the Ghost would begin moving towards the Pac-Man. If the Ghost was in a "Chase" state, it still moved at its constant speed, meaning that if the participant responded quickly, they could turn and safely exit the trial. If the Ghost entered the "Attack" state, the speed was increased such that even if the participant quickly tried to exit the trial, they would be caught by the Ghost. Please see below for a visual description of the Attack trials.

Critically, the probability of a Ghost Attack was directly determined by a beta distribution over the distance between Pac-Man and the Ghost, such that Attacks when the Pac-Man was far away from the Ghost were rare, while Attacks when Pac-Man was close were common. Whether the Ghost Attack was a Chase or Strike was randomized. Participants were directly instructed that the distance to the Ghost determined the probability of an attack. Furthermore, they were instructed that, to get the highest possible score, they would need to "balance the risks of being caught by the ghost and the rewards of getting as many dots as possible." 20% of trials are conflict-free, where the participant is free to collect the 5 dots without threat of the Ghost. On these trials, the Ghost was not present on the corridor, though participants were still instructed that they had to return to exit the trial at the side of the corridor on which they began the trial.

Additionally, the game itself was composed of 240 trials. The task was played as a series of minigames, which end when either (1) Pac-Man loses all three lives or (2) a block of twenty trials is completed. At the end of each minigame, the score was reset, and the patient was prompted to begin a new game. Direction, reward, location of the ghost, location of Pac-Man, and the ghost's starting direction are counterbalanced across trials. Patients were shown and read instructions prior to playing a short practice and were explicitly told that (1) the closer they got to the ghost the higher the chance of attack and (2) to get the highest possible score, they would need to "balance the risks of being caught by the ghost and the rewards of getting as many dots as possible". A full experimental run typically lasted approximately 20–25 min. Stimulus presentation was operated by JavaScript running in a Chrome browser.

### Data acquisition

Electrophysiological data were recorded using BCI2000, an open-source software, at each clinical site[76]. This system synchronized the Pac-Man task with LFPs, eye tracking, and behavior (Pac-Man movement) in a single data stream, which supports easy data pooling across sites and data sharing with the community[77]. The sampling rate at Barnes-Jewish Hospital was 2000 Hz, while at Loma Linda the sampling rate was 512 Hz.

### Anatomical reconstructions

We used an anatomical data processing pipeline to localize electrodes from a pre-implantation MRI and a post-implantation CT scan[78]. The MRI and CT images were aligned to a common coordinate system and fused with each other using a rigid body transformation. We then compensated for brain shifts caused by the implantation surgery. A hull of the patient's brain was generated using the Freesurfer analysis suite[79]. Electrodes were then classified by a neurologist according to the anatomical location within each patient's anatomical space. Only electrodes confirmed to be in the regions of interest were included in the analysis. For illustration purposes only, we converted patient-space electrodes into Montreal Neurological Institute (MNI) coordinates using volume-based normalization. Figure 1f shows all the electrodes used in the analysis and was created using the Python module nilearn[80].

### Data preprocessing

Offline, continuous data were low-pass filtered at 150 Hz, and notched filtered at 60 Hz and its harmonics up to 120 Hz to remove line noise. Electrodes were re-referenced using a bipolar reference scheme. We visually identified and removed channels with excessive noise or with poor contact, and each dataset was visually inspected by a neurologist to remove electrodes exhibiting epileptic spiking activity and epochs where spiking spread from the primary epileptic site. Data were epoched based on the time point of interest, either trial onset, turnaround choice, or trial end. Oscillatory and high-frequency activity was extracted at these timepoints using Morlet wavelets, where the power

of 80 log-spaced frequencies ranging from 1 to 150 was calculated using an increasing number of cycles, specifically 80 log-spaced cycles ranging from 2 to 30. Data preprocessing was carried out in Python using the MNE package[81].

## Power estimates

Theta-band and high-frequency power estimates were extracted from the TFR for all statistical analyses. Power was log-transformed, and each specific frequency was z-scored using an ITI baseline. Each frequency between 3 and 8 Hz was then averaged, resulting in a theta-band power estimate. For the high-frequency activity estimate, we followed a similar approach but averaged the individual frequencies in 20 Hz overlapping sub-bands (e.g., 70–90 Hz, 80–100 Hz, … 130–150 Hz) before averaging the power in each sub-band for a final estimate of high-frequency activity. We used a sub-band approach to minimize the effect of 1/f power scaling, which biases the power estimates to the lower frequencies in a large band, such as estimated from 70 to 150 Hz[41]. As the two hospitals had different sampling rates (2000 Hz and 512 Hz), the data were downsampled to common rate (20 Hz) after power was estimated.

For the cross-correlation analysis, we needed both power and phase information within the theta-band.

We therefore recalculated theta by bandpass filtering between 3 and 8 Hz using a zero-phase FIR filter with a Hamming window on all electrode pairs with elevated theta coherence. Data were z-scored to the ITI, time-locked to movement-onset, detrended, and only data during approach were included in this analysis.

## Connectivity measures—coherence and synchrony

Data epoched to the turnaround choice were extracted between 2500 milliseconds before and after the turnaround using Python-MNE[81]. Coherence measures were computed over time but across trials for every combination of electrodes across the regions of interest using the package MNE-connectivity. Data were resampled to 100 Hz to speed computation time. Spectrum estimation was done using Morlet Wavelets using the same parameters as above, but only within the theta-band (3–8 Hz). Three different coherence measures were calculated to ensure our results were not biased by the choice of metric: imaginary coherence, pairwise phase consistency, and the debiased estimator of the squared, weighted Phase Lag Index. To identify electrodes with a significantly elevated level of theta coherence, the true coherence metrics were compared, timepoint-by-timepoint, to a shuffled null distribution where the trials in one region were shuffled 1000 times. P values were calculated across timepoints and FDR-corrected to account for multiple comparisons. Electrode pairs were included in subsequent analyses if there were 100 milliseconds with significantly elevated coherence. Note, for imaginary coherence, we calculated the absolute value of the imaginary coherence before calculating the $P$ values.

As trial-level estimates of coherence were unstable when testing for associations with trial-level behavior, we used the correlation between two electrodes' power envelopes. For this analysis, we time-locked theta power to the final decision to avoid and filtered out the beginning of the trial when Pac-Man was stationary. We then calculated the Pearson correlation trial-by-trial for every combination of electrodes across the regions of interest. We applied this approach for both the theta-band power and for high-frequency activity estimates.

## Spectral Granger analysis

Spectral Granger causality (GC) quantifies the directionality of connectivity within specific frequency bands. Specifically, we used a State-Space formulation of Granger Causality as implemented in Python-MNE[81], which can be thought of as a more versatile and robust formulation to previous parametric approaches[39]. To improve computational efficiency, data were resampled to 100 Hz. Frequency estimation

between 3 and 8 Hz was performed using Morlet wavelets with the same parameters as above. Spectral Granger causality scores were computed using 12 lags, which in spectral State-Space GC acts as a spectral smoothing parameter across the frequencies. Granger values were then averaged across the 3–8 Hz frequency range. Following the approach of Haufe and colleagues[40] we additionally contrasted the scores with net GC scores calculated on the reversed-times series, as flipping the direction of the signal should similarly flip the estimate of directed connectivity. Thus, our final net spectral Granger scores were calculated as:

$$\text{Net Granger}(signal_a, signal_b) = \text{GC}(a \rightarrow b) -$$
$$\text{GC}(b \rightarrow a) - \text{GC}_{\text{reversed}}(a \rightarrow b) - \text{GC}_{\text{reversed}}(b \rightarrow a) \quad (1)$$

Net Granger scores were calculated for each timepoint in the final 2500 ms before the decision to avoid in all electrode pairs with significant levels of theta coherence. Only timepoints in the last 1500 ms of approach were analyzed to avoid edge artifacts. These timepoints were compared to a permuted null distribution where trial labels from one region were shuffled 1000 times. The absolute value of the Net Granger scores was used to calculate permuted $p$ values compared to this distribution. We then FDR-corrected for the number of timepoints and included electrode pairs in our subsequent analysis with 500 ms of significantly elevated net Granger scores. The net Granger scores from these electrode pairs were then fed into our directionality tests (see below).

## Cross-correlation analysis

To further validate our directionality findings, we additionally conducted a cross-correlation analysis within the theta-band. After bandpass filtering in the theta-band (see Power Estimation for details), we then concatenated all approach windows across the trials into one sequence per pair of electrodes. We next calculated a cross-correlation function with 25 lags corresponding to 250 ms on the concatenated theta signal. We then created a permuted distribution by shuffling the trial labels 200 times, while keeping the trial lengths constant to ensure that the discontinuities in the true concatenated signal were equivalent in each of the permuted samples. To assess if there was a significant correlation between two electrodes, we compared the correlation value from the true pair to the permuted distribution. If the correlation was in the top 5% of the permuted distribution, its best lag was included in our directionality test (see below).

## Bayesian linear mixed effects models

Effects were assessed using Bayesian mixed effects models that always included hierarchically grouped random effects of patient and electrodes. The benefit of these models is that they provide a single test of our hypotheses, eliminating the need to correct for the number of electrodes or develop second-level tests of how results varied across patients, while appropriately accounting for the correlations within electrodes and patients. This hierarchical random effects structure meant fitting models at times with over 3000 random effect levels. Bayesian models fit with MCMC sampling allowed us to appropriately and effectively fit this complex model structure.

Across all the Bayesian mixed effects models, we used a standard set of weakly informative priors. For intercepts, the prior was set to the normal distribution, centered on 0, with a standard deviation of 5. For beta coefficients, the prior was set to the normal distribution, centered on 0, with a standard deviation of 2. For the standard deviation of the random effects, we used an exponential distribution with a positive arrival rate of 1. Finally, we set a prior on the correlation between random effects using the

Lewandowski-Kurowicka-Joe distribution with the shape parameter of 2. All models used 4 chains and were initially fit using 5000 iterations, of which 1000 were used as warmup. If there were

indications that the models did not converge, we increased the number of iterations to a max of 10,000 iterations, of which 5000 were warmup. Rhat metrics, effective sample size, and trace plots were used to confirm model convergence and high resolution; all Rhat values for estimates were equal to or below 1.01, and all effects of interest had Rhat values equal to 1.0. Effective samples were all greater than 1000. Posterior predictive checks on the mean and standard deviation were completed across all models (See Supplementary Tables 3, 4 and 7). All models were fit using the package "brms" in R[82].

### Directionality analyses

To assess whether net Granger causality values or cross-correlation time exhibited a consistent directional trend across patients, we employed a hierarchical Bayesian model using the brms package in R. The model estimated the central tendency of net Granger values while accounting for patient- and electrode-level variability. We used weakly informative priors (see above for details). To quantify the probability that net Granger values or time lags were consistently positive across patients, we computed the Probability Positive (P+), defined as the proportion of posterior samples where the estimated effect exceeded zero. This metric provides an intuitive, probabilistic measure of directional certainty, with P+ values closer to 1 indicating strong evidence for consistent directionality.

### Reporting summary

Further information on research design is available in the Nature Portfolio Reporting Summary linked to this article.

## Data availability

The cleaned, minimally-processed, patient-level data generated in this study have been deposited in the Zenodo database under accession (https://zenodo.org/records/17726565). The data used to generate each figure in this study are provided in the Source Data file. Source data are provided with this paper.

## Code availability

Original code is deposited on Zenodo and for public download. Original code is comprised of two GitHub repos: Staveland_et_al_Pacman_Neural_Analyses (https://doi.org/10.5281/zenodo.17727554) and Staveland_et_al_Pacman_Statistics_and_Behavior (https://doi.org/10.5281/zenodo.17727552).

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

## Acknowledgements
We thank the participants for generously sharing their time and effort. This work was supported by: NSF GRFP (BRS); NIH/NINDS R01 NS-021135 (RTK); NIH/NIBIB P41 EB-018783 (PB); NIH/NINDS R01 EB-026439 (PB); NIH/NINDS U24 NS-109103 (PB); NIH/NINDS U01 NS-108916 (PB); NIH/NIMH R01 MH-120194 (JTW); McDonnell Center for Systems Neuroscience (PB); Fondazione Neurone (PB), American Epilepsy Society (JTW), NIH/BRAIN R00 NS 115918 (ELJ).

## Author contributions
B.R.S., R.T.K, and M.H. designed the experiment; R.T.K., O.K.M., J.T.W., P.B, M.D., and J.J.L. supervised data collection; O.K.M., J.T.W., P.B, M.D., J.J.L, ELJ, MP and B.R.S. collected the data; B.R.S. analyzed the data; J.O. analyzed online behavioral task data; B.B and T.M. helped to preprocess the data; B.B., T.M., ELJ, M.H., R.T.K., and B.R.S. edited the manuscript. B.R.S, M.H., and R.T.K. interpreted the data; B.R.S. wrote the manuscript; and M.H. and R.T.K. supervised the project.

## Competing interests
The authors declare no competing interests.

## Inclusion and ethics statement
This research aligns with the Inclusion & ethical guidelines embraced by Nature Communications.
