## [Transparent Peer Review file · Nature Communications]

Cortical-limbic circuit dynamics of approach-avoidance conflict in humans

Corresponding Author: Dr Brooke Staveland

Version 0:

Reviewer comments:

Reviewer #1

(Remarks to the Author)

Summary of the Article:

This study investigates the neural circuit dynamics underlying approach-avoidance conflict in humans using intracranial EEG (iEEG). Participants performed a Pac-Man-inspired task where they balanced reward collection against the threat of ghost attacks. The authors identified a distributed prefrontal-limbic circuit mediated by theta oscillations (3–8 Hz) in the hippocampus, amygdala, orbitofrontal cortex (OFC), and anterior cingulate cortex (ACC), with theta power increasing during approach and dropping during avoidance. Theta-band connectivity within this circuit correlated with longer approach times, while high-frequency activity (HFA) in the middle frontal gyrus (MFG) tracked imminent threat. The findings highlight the role of theta synchrony in coordinating approach-avoidance behavior and suggest distinct contributions of limbic and prefrontal regions to conflict processing.

Comments and Suggestions for Improvement:

Anxiety and Uncertainty Context: The introduction and discussion could better contextualize anxiety. Anxiety is a very complex symptom that involves several areas related to the anticipation of adverse or uncertain events. While the avoidance of these situations can be damaging or excessively persistent—making it a key component of anxiety—there are also other critical components, such as: The perception or management of uncertainty in situations where avoidance is not possible. Alterations in learning and the acquisition of statistical contingencies in the environment, where, again, uncertainty reflects a key aspect. In this context, the introduction and discussion should better frame these aspects. In some parts of the writing, the relationship between avoidance and anxiety appears somewhat reductionist.

The MFG's theta dynamics (Figure 2C-D) appear temporally nuanced, with an initial rise followed by a drop. A cluster-based permutation test might better capture this temporal pattern than the current 1.5 s window averaging. Clarify the rationale for the chosen time windows and reference periods in the analysis.

The contrast labels in Figure 2 (e.g., "Approach" vs. "Avoidance") are confusing. Ensure the axes and legends explicitly define the compared conditions, especially for control trials where "approach" and "avoidance" windows need clearer operational definitions.

Several critical task parameters (e.g., the distinction between pacman and ghost, ghost attack, and trial types) are presented in a fragmented manner across the manuscript. For instance, the exclusion criterion for attack trials (line 378) is introduced after the corresponding analysis in Figure 2, which may hinder interpretation.

I recommend:

Consolidating all task specifications (including attack trial mechanics and exclusion criteria) in a dedicated subsection before the results

Maintaining consistent terminology between behavioral and neural analyses

Adding a schematic timeline of trial events to Figures

Explicitly stating in each analysis section which trial types were included/excluded

This reorganization would prevent readers from needing to cross-reference multiple sections to reconstruct the experimental design, particularly for time-sensitive neural analyses where trial selection criteria directly impact interpretation. The current structure risks important methodological details being overlooked until the Discussion.

Connectivity Analysis:

The connectivity section reads as exploratory but aligns with hypotheses. Restructure to emphasize hypothesis-driven comparisons (e.g., limbic-limbic vs. limbic-MFG connectivity).

While the rationale for using Bayesian mixed modeling in connectivity analyses is understood ["Bayesian mixed-effects model within each region to predict theta coherence, using a four-level factor representing the remaining regions as the predictor."], its evaluation is impossible without clear specification in the Methods section (especially since code isn't available at this stage). I recommend explicitly detailing the applied model.

For Bayesian "purists", this statement is correct: "This analysis can be thought of as a frequentist; however, because the entire model is estimated at once, pairwise differences in coherence strength can be assessed directly without the need for post-hoc tests". However, to strengthen result robustness and better assess significance levels, I suggest additional metric:

Posterior probability thresholds (e.g., $P+ > 0.95$. as indicated in some analysis)
or Markov Chain Monte Carlo-derived p-values (p_MCMC)
Bayes Factors for key comparisons
Effective sample sizes

Directionality of Correlations:

However, caution is warranted when claiming that theta-gamma power correlations predict approach times, as correlation cannot guarantee directionality. In fact, the authors' interpretations appear to suggest the opposite: increased power correlations may instead result from escalating risk/conflict as approach time lengthens.

While the positive association between theta-band synchrony and approach duration (Fig. 4) aligns with theta's putative role in risk assessment, there are caveats that merit discussion:

Directionality Ambiguity: The correlational nature of the analysis precludes causal claims. The observed effects could reflect either:

Top-down control: Theta synchrony prolongs the approach to optimize reward-threat tradeoffs

Bottom-up drive: Accumulating risk/conflict (as proximity increases) enhances theta coupling

The authors explain that there is a theta drop similar to avoidance. This result is interesting and strange—the hypothesis would be that limbic activity relates to the probability of being attacked, but it seems to also have something to do with the subjects' "spatial" behavior. [Transition from decision-making (approach/avoidance conflict) to action execution (escape navigation) or Shift from cognitive control (OFC/ACC theta) to motor planning (parietal beta/gamma), as seen in spatial tasks]

The authors frame it in terms of proximal vs. distal threat, linking it to anxiety literature. But it's still odd in this context that hippocampal, amygdala, and OFC activity would decrease when facing a greater threat, even if it's distal. I would like to see more discussion on this.

In this context, literature on uncertainty and decision timing could help reconcile these results. For example, delta/theta band activity and neuronal spikes in the ACC are implicated in estimating and re-estimating ambiguity as a type of uncertainty [10.1371/journal.pbio.3002452, 10.1038/s41467-017-00072-y].

In such a context, a computational cognitive model could dissect variables like threat probability or distance, strengthening mechanistic claims. Address this as a future direction.

Causal Claims:

The conclusion's phrasing ("We provide evidence for a dynamic cortico-limbic network supporting approach and avoidance") overstates causal inferences. Qualify as "correlational evidence" unless causal methods (e.g., stimulation) are used.

Electrode Reporting:

Explicitly state the proportion of electrodes showing significant effects per region in all analysis (e.g., "39/127 MFG contacts exhibited HFA increases"). This avoids selection bias and clarifies regional heterogeneity.

Figure 5F:

Include amygdala data in Figure 5F for completeness, given its role in threat processing.

MFG Theta and Conflict Anticipation:

In the theta discussion, it would be interesting to consider MFG activity as reflecting conflict/danger expectation. For example, this paper [10.1038/s41467-024-54244-8] shows how MFG theta activity causally encodes conflict probability. This fits well with what was found here, especially with the high gamma activity that was the only cortical signal showing a relationship with conflict/damage probability, considering both the approach/avoidance analyses and the ghost attack analyses.

Minor Edits:

Clarify the "avoidance window" definition in control trials (line 183).

Label chase/strike trials in Figure 5A for consistency.

Check axis labels in time-frequency plots to avoid confusion (e.g., negative times = approach).

Reviewer #2

(Remarks to the Author)

In this manuscript, the authors define and investigate the role of a distributed prefrontal-limbic brain network in approach avoidance conflict. They use stereo EEG (sEEG) recorded from 20 epilepsy patients undergoing invasive monitoring for seizure localization while they performed a continuous-choice, approach-avoidance conflict decision-making task inspired by the arcade game Pac-Man. In this task, patients trade off rewards against potential losses from attack by a ghost. They show 1) Increased theta power in the hippocampus, amygdala, orbitofrontal cortex (OFC) and anterior cingulate cortex (ACC), as patients approached increasing rewards and threats, and a subsequent drop during avoidance; 2) An increase in theta band connectivity within the limbic circuit and with the medial frontal gyrus (MFG) during approach and decrease during avoidance; 3) Directional connectivity of theta from the amygdala and MFG to OFC and ACC, 4) degree of network connectivity predicted how long patients approach, with enhanced network synchronicity extending approach times and 5) when threat is imminent while being attacked by the ghost, the system dynamically switches to a sustained increase in high-frequency activity (70-150Hz) in the MFG, tracking the degree of threat. The results from this work provide evidence for a dynamic limbic-prefrontal cortical network underlying approach avoidance conflict processing in human subjects. Overall, this is a very well written article with clear description of methods and hypotheses being tested for each result. The findings in my opinion are significant in mapping a frontal-limbic brain circuit that is involved in approach avoidance conflict which can further be used to design interventions in patient population who have difficulty with resolving such conflict. Examples of such population as provided by the authors would be ones with anxiety disorders. However, information on such symptoms in their studied cohort of 20 patients with epilepsy is missing. A considerable population of patients with intractable epilepsy have comorbid anxiety disorders, but its not clear if any of the participants also had anxiety disorder and how that was factored in the analysis. As such, there is no information on neuropsychological characterization of the studied cohort. Another moderate concern is that it wasn't clear to me why the authors chose pairwise GC instead of assessing directional connectivity on the whole network such as using a partial directed coherence or directed transfer function that uses a multivariate model.

Some other minor concerns/comments are the following:

- 1) The paper title can be a bit more specific
- 2) On line 286, I believe its Fig 3e and not 3g
- 3) In the methods section (line 610), it wasn't explained why the high frequency power was calculated as averages of overlapping sub bands instead of the full band.
- 4) The zenodo link where code will be available was missing.

Reviewer #3

(Remarks to the Author)

The manuscript „Circuit dynamics of approach-avoidance conflict in humans” by Staveland and colleagues used intracranial EEG recordings from N=20 patients to define a prefrontal-limbic network, comprising of Amygdala, Hippocampus, OFC ACC, and MFG, that is engaged during approach-avoidance conflicts. The authors report distinct modulations of theta and HFA power during approach and avoidance, increased theta coherence during approach and decreased theta power during avoidance. The degree of network connectivity positively predicted approach time. Furthermore, high frequency activity in the MFG was observed during imminent threat. These results help to further our understanding about the neuronal basics underlying approach-avoidance in humans.

The manuscript is well written and the methods used are sound given some few clarifications. Accordingly, I have only minor

comments that the authors can hopefully answer.

- Please clarify whether the movement of Pac-Man and the ghost was continuous (i.e. as long as the subject pressed a key) or stepwise (i.e. move a certain distance every time a key was pressed? This is not clear from the current manuscript.
- The fact that the ghost was moving back and forth at the opposite end of the corridor is not mentioned in text, only in caption of figure 1. Could participants synchronize their movements of Pac-Man with those of the ghost? For instance, collect dots when the ghost moves away from the center and retreating when it returned towards the center – or would such a retreat have ended a trial?
- In the 20% conflict-free trials, was there no ghost or did the ghost simply not attack? If the latter, how was it obvious for the subjects that these trials were conflict-free?
- Figure 1c: I see the variability within participants but where do I see the variability across participants?
- Why was no statistical test for the different turnaround distances for large and small rewards performed?
- Significant theta-band coherence was defined as showing at least 100ms of coherence above the shuffled distribution. Can the authors explain why precisely 100ms? Does it matter whether the 100ms were before or after the decision to avoid?
- Regarding the Bayesian model on theta coherence (line 224): I understand that for each region one coherence value was extracted per subject and predicted by the coherence from the remaining four regions. Please provide information whether this was the average coherence over the whole-time window (-1500ms to +1500ms) or something else.
- Line 279: I assume this sentence refers to the theta power in the MFG? This is, however, not explicitly mentioned and not immediately clear.
- For the net Granger score, 200 permutations of the trial labels were used whereas for the theta coherence 1000 permutations were used. Could the authors clarify the rationale for this discrepancy in the number of permutations? Furthermore, was the permutation procedure applied separately for each electrode pair, or was a common null distribution computed against which all net Granger values were compared?
- Pairwise synchrony during approach: I understand that the correlation between envelopes of the theta signal of two electrodes was used as connectivity marker. However, the rest of the paragraph lacks sufficient detail. It is not mentioned that values were z-scored, which is only evident when looking at figure 4c. Furthermore, the authors mention Bayesian mixed model (I assume with time as predictor) but show no test statistics, only that "... coordinated theta activity within numerous region pairs in this circuit correlated with approach times." Can the authors please provide a more detailed description of the analysis and results?
- Line 348: Figure 4a-b shows the result from the Granger causality analysis, not the correlation with approach times.
- Colors for Amygdala and MFG in figures makes it hard to differentiate between the two.
- Given the different sampling rates between the two hospitals: Were the iEEG data down sampled to a common sampling rate?

Christian Wienke & Marcus Grüschow

Reviewer #4

(Remarks to the Author)

Version 1:

Reviewer comments:

Reviewer #1

(Remarks to the Author)

The changes address all reviewer comments and have improved the overall clarity and flow of the paper. The expanded discussion of anxiety, the clearer methodological structure, and the updated figures make the study easier to follow and more compelling. We also appreciate the addition of the analysis guide and the clearer explanation of the connectivity results. It has been a pleasure to see the manuscript evolve through the review process, and we believe the final version will

make a valuable contribution to the field and encourage further research on the neural mechanisms of approach–avoidance behavior in humans.

With best regards,
Pablo Billeke

(Remarks on code availability)

Reviewer #2

(Remarks to the Author)

Authors have satisfactorily addressed my and other reviewers' concerns raised with detailed response to each concern. I do not have any additional concerns

(Remarks on code availability)

I have partially browsed through the code . It is accessible and well documented.

Reviewer #3

(Remarks to the Author)

All our concerns have been fully addressed. We commend the authors for their efforts and an impressive paper.

(Remarks on code availability)

Reviewer #4

(Remarks to the Author)

(Remarks on code availability)

Reviewer #1 (Remarks to the Author):

Summary of the Article:

This study investigates the neural circuit dynamics underlying approach-avoidance conflict in humans using intracranial EEG (iEEG). Participants performed a Pac-Man-inspired task where they balanced reward collection against the threat of ghost attacks. The authors identified a distributed prefrontal-limbic circuit mediated by theta oscillations (3–8 Hz) in the hippocampus, amygdala, orbitofrontal cortex (OFC), and anterior cingulate cortex (ACC), with theta power increasing during approach and dropping during avoidance. Theta-band connectivity within this circuit correlated with longer approach times, while high-frequency activity (HFA) in the middle frontal gyrus (MFG) tracked imminent threat. The findings highlight the role of theta synchrony in coordinating approach-avoidance behavior and suggest distinct contributions of limbic and prefrontal regions to conflict processing.

Comments and Suggestions for Improvement:

RIC1: *The introduction and discussion could better contextualize anxiety. Anxiety is a very complex symptom that involves several areas related to the anticipation of adverse or uncertain events. While the avoidance of these situations can be damaging or excessively persistent—making it a key component of anxiety—there are also other critical components, such as: The perception or management of uncertainty in situations where avoidance is not possible.*

Alterations in learning and the acquisition of statistical contingencies in the environment, where, again, uncertainty reflects a key aspect. In this context, the introduction and discussion should better frame these aspects. In some parts of the writing, the relationship between avoidance and anxiety appears somewhat reductionist.

Response: We thank the reviewer for highlighting this issue. Anxiety is indeed a complex and multifaceted construct, and we do not intend to imply that either clinical or ‘healthy’ anxiety could be reduced to avoidance decisions. We have made edits to both the introduction and the discussion to better highlight the complexity of anxiety, and to highlight the value of using approach-avoidance paradigms as a tool for *eliciting* anxiety-related behavior, rather than being a subcomponent or reductionist model of anxiety behavior. We have especially highlighted that optimal approach-avoidance decisions require healthy functioning of systems that assess and learn about uncertainty and future threats.

Specifically, we have altered the first paragraph of the introduction as follows with changes in yellow:

“Anxiety is a complex response to ambiguous and potentially negative future states. Anxiety comprises threat appraisal and uncertainty management, among other constituents, and dysfunction in any of these domains could result in persistent or pathological anxiety. A key behavioral paradigm for investigating both clinical and normative anxiety is approach-avoidance conflict. Approach-avoidance conflict

describes the daily situations where a single choice entails potential gains and losses simultaneously, which reliably induces anxiety in everyday life (Dymond and Roche, 2009; Kirlic et al., 2017). For example, consider the decision to approach someone new while attending a party. The same action could result in the positive outcome of a nice conversation or a new friend, or a negative outcome such as social rejection or embarrassment. While anxiety in this context is normative, when avoidance becomes excessive (never speaking to anyone new, or avoiding social gatherings altogether) it becomes maladaptive (American Psychiatric Association, 2022; Dymond, 2019; Dymond and Roche, 2009). Approach-avoidance decisions are dysregulated in generalized anxiety disorder (GAD), post-traumatic stress disorder (PTSD), and agoraphobia. Specifically, in clinical disorders there is maladaptive avoidance of potentially aversive cues at the cost of large rewards (Ball and Gunaydin, 2022; Kirlic et al., 2017; Salters-Pedneault et al., 2004) and this can be disambiguated from adaptive avoidance, such as avoiding cues that are likely to result in large costs, with low probability of reward (e.g., avoiding speaking to a known rude or unkind person at a party). Understanding the cortical and limbic neural circuits that regulate healthy and adaptive approach-avoidance conflict is key to understanding the neural underpinnings of both normal and excessive clinical anxiety.”

We also introduce a discussion of possible roles that theta oscillations may be playing under approach-avoidance conflict on lines 540-545 as follows:

‘However, while there is converging evidence that theta oscillations provide cross-region mechanism for neural communication and that theta oscillations are prominent during approach-avoidance decision-making, there are open hypotheses about what kind of information is being coordinated across regions via theta oscillations. Approach-avoidance behavior is dependent on a multitude of functions, such as threat detection, uncertainty estimation, and conflict resolution between competing goals (Bryant and Barker, 2020; Ito and Lee, 2016).’

R1C2: *The MFG’s theta dynamics (Figure 2C-D) appear temporally nuanced, with an initial rise followed by a drop. A cluster-based permutation test might better capture this temporal pattern than the current 1.5 s window averaging. Clarify the rationale for the chosen time windows and reference periods in the analysis.*

Response: We thank the reviewer for noting the nuanced temporal profile of MFG theta in Figure 2. In the TFRs, there appears to be an early rise in power (–1500 to –1000ms relative to avoidance decision; see the MFG TFR in the top panel in the figure below), but supplemental analyses (now included as Supplemental Fig. 3) show that this transient broadband increase (0–16 Hz) is due to an event-related potential (ERP) at trial onset, rather than sustained theta oscillations. The ERP terminates ~1 s after trial onset, before movement begins, as can be seen in the figure on the below.

To avoid contamination of approach/avoidance comparisons by this non-oscillatory activity, our original analysis excluded the initial stationary period after trial onset, and we have now clarified this in multiple areas of the manuscript. Across participants, movement began on average 1.02 ± 0.46 s after trial onset, so we averaged theta power over the subsequent 1.5 s of approach and first 1.5 s of avoidance. This window captures sustained movement-related activity, consistent with our hypotheses, and omits the transient ERP-driven change.

Supplemental Figure 3: Time-frequency plots comparing app-avd., trial onset, and movement onset activity

Time-frequency plots, time-locked to choice to avoid (Fig. 2a, c plots)

Time-frequency plots, time-locked to trial onset

Time-frequency plots, time-locked to movement onset

Legend: Time-frequency plots of neural activity in each region in conflict trials time-locked to three different trial events: (top) the final choice to stop approaching and begin avoiding, (middle) trial onset, and (bottom) movement onset. Red (blue) indicates increases (decreases) in power. Power was log-transformed and z-scored to the mean power across the time window. It appears that MFG has increased theta power at the beginning of the approach period while time-locking to the decision to avoid. However, this is non-oscillatory, event-related activity, rather than an increase in theta oscillations.

While non-oscillatory transients can be informative, they fall outside our specific focus on theta-band oscillations. For clarity, we have (1) added an Analysis Guide to the start of the Methods, summarizing model specifications and exclusion criteria, (2) included Supplemental Fig. 3 to show the ERP-driven increase, and (3) clarified in the Results and Methods the rationale for the chosen time windows and exclusion of the stationary period. These changes are detailed below:

1. In response to this comment, as well as others below, we have now added an Analysis Guide to the beginning of the Methods sections (lines 684-686). This guide includes each major finding discussed in the manuscript and clearly outlines important information such as, the model specifications, number of models estimated, and the exclusion criteria. The relevant rows for this comment from this guide are included below:

Finding	Dependent Variable	Fixed Effect	Random Effect	Number of Estimated Models	Number of Observations	Number of Levels	Exclusion Criteria	Figures/ Tables
Theta power is higher during approach compared to avoidance during conflict trials	Mean theta power during the 1500ms before/after turning towards the exit	2-level factor, specifying avoidance (after turn) or approach (before turn)	subject/ electrode	5, one for each region	13335 - 30665	15-20/73-167	Excluded conflict-free trials, time periods before the Pac-Man initiated movement; trials where 0 dots were collected; trials where Ghost attacked	Figure 2; Extended Data Table 1
Theta power is not higher during approach compared to reward during conflict-free trials	Mean theta power during the 1500ms before/after turning towards the exit	2-level factor, specifying return (after turn) or approach (before turn)	subject/ electrode	5, one for each region	4718 - 11146	15-20/73-167	Excluded conflict trials, time periods before the Pac-Man initiated movement; trials where 0 dots were collected	Supplemental Figure 2; Extended Data Table 1

2. The figure in this response is included in the supplement as Supplemental Figure 3. We direct the reader to this figure on lines 247-248:

“see Supplemental Figure 3 for an analysis of the brief increase in low frequency power from an event-related potential at the beginning of the approach period in the MFG contacts.”

3. We have added text to explain the choice to only include movement-related behavior on lines 213-218:

“We next tested if the approach period correlated with greater theta power compared to the avoidance period across the different regions. We excluded the initial period of the trial where the Pac-Man was stationary to best compare the approach activity to avoidance activity. On average, patients began moving 1.02+/- 0.46 seconds after trial onset and approached both the threat (Ghost) and reward (dots) for 1.39 +/- 0.22

seconds. We therefore averaged theta power during the final 1.5 seconds of approach and first 1.5 seconds of avoidance in each electrode and patient, on trials without a Ghost Attack. We estimated a Bayesian mixed effects model of theta power using their approach/avoidance decisions...”

R1C3: *The contrast labels in Figure 2 (e.g., "Approach" vs. "Avoidance") are confusing. Ensure the axes and legends explicitly define the compared conditions, especially for control trials where "approach" and "avoidance" windows need clearer operational definitions.*

Response: We thank the reviewer for identifying this point of confusion and have taken multiple steps to clarify the time windows used in conflict and conflict-free trials.

First, we made changes to the text:

- On lines 228-232: “In the subset of control trials that were conflict-free, i.e. where patients collected reward without threat of the ghost, we ran an equivalent model, where the approach period (e.g., 1.5 seconds before the decision to turn around, made after all rewards were collected) was compared against the 1.5 seconds after the decision to turn. We did not find a significant difference in theta power between the approach and return windows...”

Second, we made changes to Figure 2. We added a small task schematic, which identifies the approach and avoidance periods. We use “Return” to identify the period in Conflict-free trials after the final decision to turn back, which corresponds with the “Avoid” window in the Conflict trials, and now highlight the lack of ghost with a small symbol. You can review the changes to Figure 2 on the next page.

Finally, in the Figure 2 Legend we made the following adjustments for clarity:

Fig. 2 a Time-frequency analysis of neural activity in conflict trials time-locked to the final choice to stop approaching and begin avoiding across all electrodes in the limbic regions (hippocampus, amygdala, OFC and ACC). Red (blue) indicates increases (decreases) in power. Power was log-transformed and z-scored to the mean power across the 3-second time window. Dotted box indicates the theta band (3-8Hz) See Supplemental Figure 2 for time-frequency analysis of the conflict-free trials. **b** Average theta power in the 1500ms before (called “Approach”) and after the choice to turn around on each trial (called “Avoid” or “Return”) for each limbic region in conflict-trials (ghost present) or conflict-free trials (ghost absent). Dots indicate average theta power in electrodes across trials, triangles indicate average theta power across electrodes within a participant. Significance was assessed using linear-mixed effects models with random effects of participant and electrode. Stars indicate that the 95% credible interval did not include 0. **c** Time-frequency analysis of the conflict trials and theta-power analysis for the MFG....

Fig. 2: Theta power is elevated during approach in limbic regions, but not MFG, in a real-time app.-avd. conflict task

a. Average limbic time-frequency plots, time-locked to choice to avoid

b. Limbic theta is only elevated during conflict-trials

c. Middle Frontal Gyrus does not show limbic theta pattern

d. Dissociation between θ theta & HFA, MFG & Limbic

RIC4: *Several critical task parameters (e.g., the distinction between pacman and ghost, ghost attack, and trial types) are presented in a fragmented manner across the manuscript. For instance, the exclusion criterion for attack trials (line 378) is introduced after the corresponding analysis in Figure 2, which may hinder interpretation.*

I recommend:

- *Consolidating all task specifications (including attack trial mechanics and exclusion criteria) in a dedicated subsection before the results*
- *Maintaining consistent terminology between behavioral and neural analyses*
- *Adding a schematic timeline of trial events to Figures*
- *Explicitly stating in each analysis section which trial types were included/excluded*

This reorganization would prevent readers from needing to cross-reference multiple sections to reconstruct the experimental design, particularly for time-sensitive neural analyses where trial selection criteria directly impact interpretation. The current structure risks important methodological details being overlooked until the Discussion.

Response: We appreciate the suggestion to consolidate the task parameters and to better link each analysis to the trial types and timepoints used. We agree with each point and have made two major changes to increase the clarity of the manuscript. First, we expanded on the description of the task in a subsection before the results, as was suggested (copied below). Second, we have added an “Analysis Guide” to beginning of the methods. Here we have listed each of the main findings, the specifications of the model, the exclusion criteria, and the corresponding figures and tables.

Here is the new Task Description (lines 130-168):

Task Design: We designed an approach-avoidance conflict task based on the arcade game Pac-Man (See Figure 1a). On each trial, Pac-Man, the Ghost, and dots are placed along a single corridor that runs horizontally across the screen. The participant controls the Pac-Man and they are instructed to collect as many dots as they can while avoiding the Ghost, which is moving back and forth at a constant speed at the opposite side of the corridor. The participant makes a single button press, using the left or right arrow keys, to initiate movement along the corridor at a constant speed. By moving towards the center of the corridor, they can collect up to 5 dots, resulting in points, which varied in either small (10 points) or large (20 point) sizes and were placed in pseudorandom configurations between Pac-Man and the Ghost. Importantly, at any point in the trial, they could also choose to move away from the center of the corridor (and, thus, away from both the dots and the Ghost) where they could exit to the next trial at the end of the corridor. If Pac-Man was “caught” by the Ghost, they lost one of their three lives, they also lost all the points acquired on that trial, and the Ghost capture was accompanied by a “death animation” used in the arcade game (e.g., a loud sound and the Pac-Man shape rotates and disappears). The Ghost’s starting direction and

location, the distance between the dots and Pac-Man, and Pac-Man's starting location are counterbalanced across trials.

The Ghost had three states, which we term "Pace", "Chase", and "Strike". The default state was "Pace", where the Ghost moves in a set path back and forth at the end of the corridor. The end of the path was a jittered distance away from the last (5th) dot. In this state, the Ghost moved at the same speed as the Pac-Man and did not deviate from its path to capture the Pac-Man. The remaining two states, which we jointly call a "Ghost Attack", were designed to motivate behavior on a typical trial (See Results for Details). On these Attack trials, the Ghost would begin moving towards the Pac-Man. If the Ghost was in a "Chase" state, it still moved at its constant speed, meaning that if the participant responded quickly, they could turn and safely exit the trial. If the Ghost entered the "Strike" state, the speed was increased such that even if the participant quickly tried to exit the trial, they would be caught by the Ghost. Please see below for a visual description of the Attack trials.

Critically, the probability of a Ghost Attack was directly determined by a beta distribution over the distance between Pac-Man and the Ghost, such that Attacks when the Pac-Man was far away from the Ghost were rare, while Attacks when Pac-Man was close were common. Whether the Ghost Attack was a Chase or Strike was randomized. Participants were directly instructed that the distance to the Ghost determined the probability of an attack. Furthermore, they were instructed that, to get the highest possible score, they would need to "balance the risks of being caught by the ghost and the rewards of getting as many dots as possible."

Finally, 20% of trials are conflict-free, where the participant is free to collect the 5 dots without threat of the Ghost. On these trials, the Ghost was not present on the corridor, though participants were still instructed that they had to return to exit the trial at the side of the corridor on which they began the trial.

And here are the first rows of the Analysis Table, for the full Table see Table 1 beginning on line 684 of the manuscript:

Finding	Dependent Variable	Fixed Effect	Random Effect	Number of Estimated Models	Number of Observations	Number of Levels	Exclusion Criteria	Figures/ Tables
Behavioral Result: If the last dot (reward) was large, participants were closer to the ghost at turnaround	distance between Pac-Man and the Ghost at the turnaround point	2-level factor, specifying if the last dot was large or small	subject	1	34298	211 (191 online participants + 20 iEEG patients)	Excluded trials where no turn was made; conflict-free trials	Figure 1d; Extended Data Table 1
Theta power is higher during approach compared to avoidance during conflict trials	Mean theta power during the 1500ms before/after turning towards the exit	2-level factor, specifying avoidance (after turn) or approach (before turn)	subject/ electrode	5, one for each region	13335 - 30665	15-20/73-167	Excluded conflict-free trials, time periods before the Pac-Man initiated movement; trials where 0 dots were collected; trials where Ghost attacked	Figure 2; Extended Data Table 1

We further address each bullet:

- “Consolidating all task specifications...”:
 - **Response:** We have done this, please see above.
- “Maintain consistent terminology”:
 - **Response:** In reworking the task description and in adding details to each analysis, we have maintained the language that we introduce in the task description.
- “Adding a schematic”
 - **Response:** In addition to the task schematics in Figure 1 and Figure 5, we have now added a small schematic to Figure 2 to better clarify the approach and avoidance windows.
- “Explicitly stating in each analysis”:
 - **Response:** We have done this, please see above response.

RIC5: *The connectivity section reads as exploratory but aligns with hypotheses. Restructure to emphasize hypothesis-driven comparisons (e.g., limbic-limbic vs. limbic-MFG connectivity).*

Response: We have revised the connectivity section to better clarify our hypothesis-driven analyses and exploratory analyses in this section. We now break the results of this section into three paragraphs: (1) evidence that *supports* our hypothesis of elevated limbic-to-limbic connectivity compared to limbic-to-MFG connectivity, (2) evidence *against* our hypothesis of elevated limbic-to-limbic connectivity compared to limbic-to-MFG connectivity, and (3) exploratory connectivity within the MFG. Here are those following sections (beginning on line 292):

“Evidence for elevated limbic-to-limbic connectivity compared to limbic-to-MFG connectivity: As predicted, we found that the amygdala exhibited higher theta

coherence with the hippocampus compared to the MFG, with an estimated connectivity strength ratio of 1.49 (95% credible interval [1.29, 1.75], $P_+ > 0.99$, $ESS_{\text{bulk}} = 7582.3$; See Figure 3c). This indicates that amygdala-hippocampus coherence was ~1.5x stronger than amygdala-MFG coherence during the approach. Similarly, we found that amygdala-OFC coherence and amygdala-ACC coherence were both stronger than amygdala-MFG coherence, with a connectivity strength ratio of 1.22 (95% CI [1.07, 1.36], $P_+ > 0.99$, $ESS_{\text{bulk}} = 6426.1$; see Supplemental Figure 5) and 1.14 (95% CI [1.03, 1.28], $P_+ > 0.99$, $ESS_{\text{bulk}} = 11399.5$; see Supplemental Figure 5), respectively. We found similar results in the hippocampus (See Figure 3c; Supplemental Figure 5; Supplemental Table 3); hippocampus coherence with the amygdala was 1.41 (95% CI [1.23, 1.65], $P_+ > 0.99$, $ESS_{\text{bulk}} = 5789.6$) times greater than its theta coherence with the MFG, and 1.14 (95% CI [1.03, 1.27], $P_+ > 0.99$, $ESS_{\text{bulk}} = 5719.2$) times greater with the OFC compared to the MFG. Finally, we found that within the OFC, there is stronger OFC-ACC connectivity compared to OFC-MFG connectivity, with a connectivity strength ratio of 1.2 (95% CI [1.07-1.36], $P_+ > 0.99$, $ESS_{\text{bulk}} = 5742.9$). These results provide evidence that the amygdala and hippocampus are more strongly connected to other limbic regions via theta compared to the MFG (See Supplemental Figure 5 for all regional comparisons using Imaginary Coherence; see Supplemental Figure 6 for results using Pairwise Phase Consistency and Phase Lag Index; see Supplemental Table 3 for full model results across the three connectivity metrics; see Supplemental Table 4 for posterior predictive checks across connectivity models; see Extended Data Table 1 for model estimates).

Evidence against elevated limbic-to-limbic connectivity compared to limbic-to-MFG connectivity: Within the ACC, we see stronger theta coherence with the MFG compared to the amygdala and hippocampus, where ACC-amygdala coherence was only 0.76 (95% CI [0.65, 0.88], $P_+ < 0.01$, $ESS_{\text{bulk}} = 4476.4$) times the ACC-MFG coherence and ACC-hippocampus was only 0.77 (95% CI [0.65, 0.90], $P_+ < 0.01$, $ESS_{\text{bulk}} = 3670.9$) times ACC-MFG coherence (See Figure 3c, Supplemental Figure 5 and Supplementary Table 3 for details). Similarly, the ACC showed preferential connectivity with the OFC compared to the subcortical limbic regions (See Figure 3c, Supplemental Figure 5 and Supplementary Table 3 for details), meaning that, while the ACC shared similar theta power modulations as the subcortical limbic regions, the ACC was preferentially connected to other prefrontal regions compared to the two subcortical limbic regions. Furthermore, while the OFC had preferential connectivity with the ACC compared to the MFG, there was no difference in connectivity strength between OFC-hippocampus and OFC-MFG connectivity, nor any difference between OFC-amygdala and OFC-MFG connectivity (See Figure 3c, Supplementary Figure 5, Supplementary Figure 6, Supplementary Table 3-4 for details). Thus, instead of finding

evidence for increased limbic-to-limbic connectivity, we found that the OFC had a more distributed connectivity pattern, while the ACC was more strongly connected to the OFC and MFG. Contrary to our initial hypothesis, the prefrontal regions, including the MFG, shared elevated theta coherence, indicating the connectivity was not limited to the limbic regions.

Connectivity within the MFG: We examined the connectivity patterns within the MFG. We found that the MFG also had preferential connectivity with the other prefrontal regions (OFC, ACC) compared to subcortical regions (See Figure 3c, Supplemental Figure 5 and Supplementary Table 3 for details), matching the profile we found in the ACC.”

R1C6: *While the rationale for using Bayesian mixed modeling in connectivity analyses is understood [“Bayesian mixed-effects model within each region to predict theta coherence, using a four-level factor representing the remaining regions as the predictor.”], its evaluation is impossible without clear specification in the Methods section (especially since code isn't available at this stage). I recommend explicitly detailing the applied model.*

Response: We thank the reviewer for the helpful comment. We addressed this comment in three ways. First, we have made an Analysis Guide (beginning on line 684) which is at the beginning of the Methods and can be used as a shorthand guide for matching a given result to its statistical model. Second, we have included the model output summaries in excel sheets as and Extended Data Table, so readers can easily view and interpret the output for themselves. Third, we have added details to the analysis description in the Results.

The relevant portion of the Analysis Guide is copied below:

Finding	Dependent Variable	Fixed Effect	Random Effect	Number of Estimated Models	Number of Observations	Number of Levels	Exclusion Criteria	Figures/ Tables
Prefrontal and subcortical regions form subnetworks within a wider theta circuit	Average theta coherence during the approach period (1500ms preceding the choice to avoid) for a given region, logged	4-level factor, specifying partner region	subject/ first electrode of pair	5, one for each region	1923 - 4097	15-20/115-355	conflict-free trials; trials where 0 dots were collected	Figure 3 c; Supplemental Figures 5 & 6; Supplemental Tables 3 &4; Extended Data Table 1

Here is the relevant change of the updated result on lines 281-285:

“We then tested if the consistent limbic theta power elevations were indicative of elevated theta coherence between limbic regions, as compared to the MFG. To test this, we ran a Bayesian mixed effects model within each region to predict the average

theta coherence during the approach window (1500ms preceding the avoidance decision), using a four-level factor representing the remaining regions as the predictor.”

RIC7: *For Bayesian "purists", this statement is correct: "This analysis can be thought of as a frequentist; however, because the entire model is estimated at once, pairwise differences in coherence strength can be assessed directly without the need for post-hoc tests". However, to strengthen result robustness and better assess significance levels, I suggest additional metric:*

- *Posterior probability thresholds (e.g., $P+ > 0.95$. as indicated in some analysis)*
- *or Markov Chain Monte Carlo-derived p-values (p_MCMC)*
- *Bayes Factors for key comparisons*
- *Effective sample sizes*

Response: We thank the reviewer for this comment, we now include (1) the posterior probability thresholds, (2) effective sample sizes, and (3) the number of electrodes that demonstrate a given effect (when appropriate), as suggested in a later comment.

- **On lines 181-183:** ‘As predicted, when the last reward in the corridor was large, participants were willing to move closer to the ghost to collect the large reward (Estimate_{small>large}: 0.11, 95% CI [0.09-0.13], $P+ > 0.99$, $BF_{10} > 100$, $ESS_{bulk} = 14861.4$ ’
- **On lines 220-228:** ‘We found that theta power was elevated during approach compared to avoidance in conflict trials within the hippocampus (See Figure 2, Panel a; Conflict-Trial Estimate_{approach>avoidance}: 0.24, 95% CI [0.10-0.37], $P+ > 0.99$, $BF_{10} > 4$, $ESS_{bulk} = 3838.9$, Ratio of Sig. Electrodes $_{P+ > 0.95} = 72/115$), amygdala (Conflict-Trial Estimate_{approach>avoidance}: 0.16, 95% CI [0.07-0.26], $P+ > 0.99$, $BF_{10} > 3$, $ESS_{bulk} = 5903.8$, Ratio of Sig. Electrodes $_{P+ > 0.95} = 59/101$), OFC (Conflict-Trial Estimate_{approach>avoidance}: 0.21, 95% CI [0.13-0.29], $P+ > 0.99$, $BF_{10} > 100$, $ESS_{bulk} = 4243.2$, Ratio of Sig. Electrodes $_{P+ > 0.95} = 127/167$) and ACC (Conflict-Trial Estimate_{approach>avoidance}: 0.26, 95% CI [0.14-0.38], $P+ > 0.99$, $BF_{10} > 18$, $ESS_{bulk} = 7222.9$, Ratio of Sig. Electrodes $_{P+ > 0.95} = 50/73$).’
- **On lines 231-237:** ‘We did not find a significant difference in theta power between the approach and return windows (Hippocampus Estimate_{approach>return}: 0.08, 95% CI [-0.08-0.25], $P+ = 0.84$, $BF_{10} = 0.07$, $ESS_{bulk} = 2778.1$, Ratio of Sig. Electrodes $_{P+ > 0.95} = 72/115$; Amygdala Estimate_{approach>return}: 0.11, 95% CI [-0.02-0.22], $P+ = 0.96$, $BF_{10} = 0.14$, $ESS_{bulk} = 4697.1$, Ratio of Sig. Electrodes $_{P+ > 0.95} = 38/101$; OFC Estimate_{approach>return}: 0.13, 95% CI [-0.01-0.27], $P+ = 0.97$, $BF_{10} > 0.21$, $ESS_{bulk} = 4350.9$, Ratio of Sig. Electrodes $_{P+ > 0.95} = 71/167$; ACC Estimate_{approach>return}: 0.12, 95% CI [-0.07-0.31], $P+ = 0.90$, $BF_{10} > 0.11$, $ESS_{bulk} = 7437.9$, Ratio of Sig. Electrodes $_{P+ > 0.95} = 24/73$)’

- **On lines 243-247:** ‘We did not find evidence for increased theta power in the approach period compared to the avoidance period in the MFG (Conflict-Trial Estimate_{approach>avoidance}: 0.06, 95% CI [-0.08-0.19], P+ = 0.82, BF₁₀ = 0.05, ESS_{bulk} = 4642.4, Ratio of Sig. Electrodes_{P+ > 0.95} = 31/143; Conflict-Free Estimate_{approach>return}: -0.06, 95% CI [-0.21-0.11], P+ = 0.24, BF₁₀ = 0.05, ESS_{bulk} = 4166.8, Ratio of Sig. Electrodes_{P+ > 0.95} = 11/143’.
- **On lines 294-295:** ‘... with an estimated connectivity strength ratio of 1.49 (95% credible interval [1.29, 1.75], P+ > 0.99 , ESS_{bulk} = 7582.3’
- **On lines 298-299:** ‘... with a connectivity strength ratio of 1.22 (95% CI [1.07, 1.36] , P+ > 0.99 , ESS_{bulk} = 6426.1; see Supplemental Figure 5) and 1.14 (95% CI [1.03, 1.28], P+ > 0.99 , ESS_{bulk} = 11399.5; see Supplemental Figure 5), respectively’
- **On lines 301-303:** ‘... hippocampus coherence with the amygdala was 1.41 (95% CI [1.23, 1.65], P+ > 0.99 , ESS_{bulk} = 5789.6) times greater than its theta coherence with the MFG, and 1.14 (95% CI [1.03, 1.27], P+ > 0.99 , ESS_{bulk} = 5719.2) times greater with the OFC compared to the MFG.’
- **On lines 304-305:** ‘... with a connectivity strength ratio of 1.2 (95% CI [1.07-1.36], P+ > 0.99 , ESS_{bulk} = 5742.9).’
- **On lines 313-317:** ‘Within the ACC, we see stronger theta coherence with the MFG compared to the amygdala and hippocampus, where ACC-amygdala coherence was only 0.76 (95% CI [0.65, 0.88], P+ < 0.01 , ESS_{bulk} = 4476.4) times the ACC-MFG coherence and ACC-hippocampus was only 0.77 (95% CI [0.65, 0.90], P+ < 0.01 , ESS_{bulk} = 3670.9) times ACC-MFG coherence.’
- **On lines 347-349:** ‘We found that average connectivity values increased during the time the patient approaches (Estimate: 0.09, 95% CI [0.05-0.14], P+ > 0.99, BF₁₀ > 9, ESS_{bulk} = 7680.4, Ratio of Sig. Electrodes_{P+ > 0.95} = 1782/2816...’
- **On lines 363-365:** ‘...theta coherence drops after the patients began avoiding and moved to exit the trial (Estimate: -0.11, 95% CI [-0.16 - -0.06], P+ < 0.01, BF₁₀ > 14, ESS_{bulk} = 3520.3, Ratio of Sig. Electrodes_{P+ > 0.95} = 1824/2816...’
- **On lines 424-426:** ‘As hypothesized, the model found an overall significant correlation between approach times and theta synchrony (Estimate: 0.07, 95% CI [0.04-0.09], P+ > 0.99, BF₁₀ > 100, ESS_{bulk} = 2816.6)’
- **On lines 439-441:** ‘While the model did not find an overall significant correlation between approach times and HFA synchrony (Estimate: -0.01, 95% CI [-0.04-0.01], P+ = 0.1, BF₁₀ = 0.01, ESS_{bulk} = 2159.4...’
- **On lines 480-481:** ‘... where right MFG contacts resulted in a significant elevation of HFA in the second after an attack began (estimate: -0.16 95% CI [-0.29 - -0.04], P+ < .01, BF₁₀ > 1.75, ESS_{bulk} = 6782.9; See Figure 5d).’

- **On lines 488-492:** As predicted, we found a main effect of trial type, where chase trials had decreased HFA compared to strike trials (Estimate: -0.16 95% CI [-0.21 - -0.12]), $P+ < .01$, $BF_{10} > 100$, $ESS_{bulk} = 13160.1$) and a significant interaction with time where chase trials showed dropping HFA as patients neared the end of the trial (estimate: -0.12 95% CI [-0.17 - -0.08], $P+ < .01$, $BF_{10} > 100$, $ESS_{bulk} = 13324.6$)...

We did not report Bayes Factors for the factor-level contrasts because in this hierarchical setting they strongly depend on the prior for contrasts between the factors, which is not explicitly parameterized in the model. This makes the Bayes Factors for these values numerically unstable and difficult to interpret. Instead, we report Probability Positive (P+), 95% confidence intervals, and effective sample sizes.

RIC8: *However, caution is warranted when claiming that theta-gamma power correlations predict approach times, as correlation cannot guarantee directionality. In fact, the authors' interpretations appear to suggest the opposite: increased power correlations may instead result from escalating risk/conflict as approach time lengthens.*

While the positive association between theta-band synchrony and approach duration (Fig. 4) aligns with theta's putative role in risk assessment, there are caveats that merit discussion:

- *Directionality Ambiguity: The correlational nature of the analysis precludes causal claims. The observed effects could reflect either:*
- *Top-down control: Theta synchrony prolongs the approach to optimize reward-threat tradeoffs*
- *Bottom-up drive: Accumulating risk/conflict (as proximity increases) enhances theta coupling*

Response: This valuable comment resulted in an expansion of the Discussion to discuss these interpretations and better clarify our results. We explicitly highlight the directional ambiguity of the theta synchrony result and elaborate on the possible mechanistic role theta oscillations might be playing during approach-avoidance conflict.

Here is the relevant portion (beginning on line 518) of the reworked Discussion:

“Low-frequency theta oscillations coordinate neural activity between distal regions (Fries, 2015, 2005; Helfrich and Knight, 2016; Phillips et al., 2014) facilitating information transfer (Canolty and Knight, 2010; Yuste, 2015). Absent between-region synchrony, inputs may arrive at random phases of the excitability cycle hindering effective inter-regional communication. Notably, the degree of theta coherence correlates with behavioral performance in a wide range of attention, learning, emotional and working-memory tasks (Backus et al., 2016; Harris and Gordon, 2015; Johnson et al., 2017; Watrous et al., 2013; Zheng et al., 2019, 2017). Our results support a key role for theta synchrony during approach-avoidance. We found many

theta-cohering pairs of electrodes across the prefrontal-subcortical circuit with theta band synchrony peaking at the final decision to avoid—when the levels of threat, conflict and the need for control were the highest. The prefrontal cortex is thought to have a particular role in coordinating the oscillatory activity underlying cognitive control and goal-directed behavior (Helfrich and Knight, 2016), and midline theta oscillations recorded from scalp EEG have been long-associated with uncertainty and cognitive control (Cavanagh et al., 2012; Cavanagh and Frank, 2014). Furthermore, in the context of approach-avoidance, Amemori et al 2024 reported that reduced connectivity in a PFC-limbic circuit results in more avoidant behavior in non-human primates (Amemori et al., 2024b). In line with these results, we found that the MFG had Granger causal influence on theta oscillations in both the OFC and ACC, and increased theta synchrony across multiple nodes in the prefrontal-limbic theta network correlated with longer approach times in the task, meaning that patients took on more risk to receive increased reward in those trials. Importantly, while Amemori focused on alpha oscillations in a restricted set of limbic regions, they defined the alpha band as between 5-13Hz, and reported peaks at 7Hz, which overlaps with our 3-8Hz theta band definition for this study.

However, while there is converging evidence that theta oscillations provide a cross-region mechanism for neural communication and that theta oscillations are prominent during approach-avoidance decision-making, there are open hypotheses about what kind of information is being coordinated across regions via theta oscillations.

Approach-avoidance behavior is dependent on a multitude of functions, such as threat detection, uncertainty estimation, and conflict resolution between competing goals (Bryant and Barker, 2020; Ito and Lee, 2016). For example, in our task increased theta oscillations and coherence could reflect increased volitional cognitive control to successfully obtain more reward in the face of increasing task demands. Alternatively, the accumulating risk and conflict over the trial could drive increasing theta synchrony in a bottom-up manner. The Ghost attack trials are informative as we observe a drop in theta power after the turn to avoid a direct attack by the Ghost, similar to the decrease in theta power during an attack-free trial. Alternatively, we could have seen theta oscillations continue or even increase as the Ghost attacks and gets closer to the Pac-Man, which would suggest a greater correspondence between risk or anxiety and theta oscillations. Instead, theta power falls approximately one second after the onset of the Ghost Attack, corresponding to after the participant has turned to avoid the attack. This is compatible with a role for theta oscillations in navigating uncertainty, conflict, or exerting cognitive control—before an attack the ghost needs to be closely monitored, but after an attack and a turn is made the need to control behavior concludes. Furthermore, our findings that theta coherence drops during the avoidance period

support the idea that theta oscillations track uncertainty or cognitive control—while there may still be conflicting goals after turning to avoid due to remaining reward in the trial, theta coherence begins dropping during this time, suggesting a decreased need to predict the ghost’s actions (uncertainty) and to adjust one’s behavior in response to those predictions (cognitive control). Computational models are needed to fully test these interpretations, but the current results are in accord with work showing a role for theta oscillations handling uncertainty in the ACC, especially when that uncertainty requires behavioral adaptation (Monosov, 2017; Valdebenito-Oyarzo et al., 2024). Given that the OFC and ACC were the main receivers of theta oscillations in our task, they may be well-suited to integrating information from the other regions in order to respond to uncertainty.”

We also adjusted any lines in the Results or Discussion that implied a specific causal direction for the theta or HFA synchrony findings:

- **On line 415:** ‘Pairwise synchrony during approach behavior correlates with choice to avoid:’
- **On lines 441-442:** ‘...we did find a more restricted set of region pairs that correlated with approach time’
- **On lines 452-453:** ‘Additionally, we have found that synchrony in both the theta band and HFA in these regions correlates with approach times and that the OFC connects the prefrontal regions with the hippocampus and amygdala.’
- **On lines 568-569:** ‘While the OFC-MFG and OFC-ACC HFA synchrony correlated with longer approach times, hippocampus-MFG HFA synchrony correlated with shorter approach times.’
- In the Figure 4 legend, sections c-e: ‘Trial-by-trial theta and HFA synchrony correlates with approach times’ & ‘Lines indicate that the connected regions correlated with approach times’

RIC9: *The authors explain that there is a theta drop similar to avoidance. This result is interesting and strange—the hypothesis would be that limbic activity relates to the probability of being attacked, but it seems to also have something to do with the subjects' "spatial" behavior. [Transition from decision-making (approach/avoidance conflict) to action execution (escape navigation) or Shift from cognitive control (OFC/ACC theta) to motor planning (parietal beta/gamma), as seen in spatial tasks]*

- *The authors frame it in terms of proximal vs. distal threat, linking it to anxiety literature. But it's still odd in this context that hippocampal, amygdala, and OFC activity would decrease when facing a greater threat, even if it's distal. I would like to see more discussion on this.*
- *In this context, literature on uncertainty and decision timing could help reconcile these results. For example, delta/theta band activity and neuronal spikes in the ACC are implicated in estimating and re-estimating ambiguity as a type of uncertainty [10.1371/journal.pbio.3002452, 10.1038/s41467-017-00072-y].*

- *In such a context, a computational cognitive model could dissect variables like threat probability or distance, strengthening mechanistic claims. Address this as a future direction.*

Response: We thank the reviewer for highlighting a finding that was underdiscussed in the initial submission. We agree that the result is interesting and worthy of further exploration, as it provides insight as to the role of theta oscillations in the task. In response, we have added lines to the Discussion (also included above) explicitly discussing this finding and how it relates to the role of theta oscillations in approach-avoidance conflict.

Here is the relevant portion of the reworked Discussion, beginning on line 540:

“However, while there is converging evidence that theta oscillations provide a cross-region mechanism for neural communication and that theta oscillations are prominent during approach-avoidance decision-making, there are open hypotheses about what kind of information is being coordinated across regions via theta oscillations. Approach-avoidance behavior is dependent on a multitude of functions, such as threat detection, uncertainty estimation, and conflict resolution between competing goals (Bryant and Barker, 2020; Ito and Lee, 2016). For example, in our task increased theta oscillations and coherence could reflect increased volitional cognitive control to successfully obtain more reward in the face of increasing task demands. Alternatively, the accumulating risk and conflict over the trial could drive increasing theta synchrony in a bottom-up manner. The Ghost attack trials are informative as we observe a drop in theta power after the turn to avoid a direct attack by the Ghost, similar to the decrease in theta power during an attack-free trial. Alternatively, we could have seen theta oscillations continue or even increase as the Ghost attacks and gets closer to the Pac-Man, which would suggest a greater correspondence between risk or anxiety and theta oscillations. Instead, theta power falls approximately one second after the onset of the Ghost Attack, corresponding to after the participant has turned to avoid the attack. This is compatible with a role for theta oscillations in navigating uncertainty, conflict, or exerting cognitive control—before an attack the ghost needs to be closely monitored, but after an attack and a turn is made the need to control behavior concludes. Furthermore, our findings that theta coherence drops during the avoidance period support the idea that theta oscillations track uncertainty or cognitive control—while there may still be conflicting goals after turning to avoid due to remaining reward in the trial, theta coherence begins dropping during this time, suggesting a decreased need to predict the ghost’s actions (uncertainty) and to adjust one’s behavior in response to those predictions (cognitive control). Computational models are needed to fully test these interpretations, but the current results are in accord with work showing a role for theta oscillations handling uncertainty in the ACC, especially when that uncertainty requires behavioral adaptation (Monosov, 2017; Valdebenito-Oyarzo et al.,

2024). Given that the OFC and ACC were the main receivers of theta oscillations in our task, they may be well-suited to integrating information from the other regions in order to respond to uncertainty.”

R1C10: *The conclusion’s phrasing (“We provide evidence for a dynamic cortico-limbic network supporting approach and avoidance”) overstates causal inferences. Qualify as “correlational evidence” unless causal methods (e.g., stimulation) are used.*

Response: We have adjusted the phrasing to better reflect the types of evidence used in the paper. We have made the following changes to line 500:

“We provide behavioral and electrophysiological evidence for a dynamic cortico-limbic network supporting approach and avoidance...”

R1C11: *Explicitly state the proportion of electrodes showing significant effects per region in all analyses (e.g., “39/127 MFG contacts exhibited HFA increases”). This avoids selection bias and clarifies regional heterogeneity.*

Response: We thank the reviewer for this suggestion. We have now added the number of significant electrodes to each of the main results. Here, we count an electrode as “significant” if that electrode’s posterior distribution had a Probability Positive (P+) value > 0.95 and have made the corresponding changes:

- **On lines 220-228:** ‘We found that theta power was elevated during approach compared to avoidance in conflict trials within the hippocampus (See Figure 2, Panel a; Conflict-Trial Estimate_{approach>avoidance}: 0.24, 95% CI [0.10-0.37], P+ > 0.99, BF₁₀ > 4, ESS_{bulk} = 3838.9, Ratio of Sig. Electrodes_{P+ > 0.95} = 72/115), amygdala (Conflict-Trial Estimate_{approach>avoidance}: 0.16, 95% CI [0.07-0.26], P+ > 0.99, BF₁₀ > 3, ESS_{bulk} = 5903.8, Ratio of Sig. Electrodes_{P+ > 0.95} = 59/101), OFC (Conflict-Trial Estimate_{approach>avoidance}: 0.21, 95% CI [0.13-0.29], P+ > 0.99, BF₁₀ > 100, ESS_{bulk} = 4243.2, Ratio of Sig. Electrodes_{P+ > 0.95} = 127/167) and ACC (Conflict-Trial Estimate_{approach>avoidance}: 0.26, 95% CI [0.14-0.38], P+ > 0.99, BF₁₀ > 18, ESS_{bulk} = 7222.9, Ratio of Sig. Electrodes_{P+ > 0.95} = 50/73).’
- **On lines 231-237:** ‘We did not find a significant difference in theta power between the approach and return windows (Hippocampus Estimate_{approach>return}: 0.08, 95% CI [-0.08-0.25], P+ = 0.84, BF₁₀ = 0.07, ESS_{bulk} = 2778.1, Ratio of Sig. Electrodes_{P+ > 0.95} = 72/115; Amygdala Estimate_{approach>return}: 0.11, 95% CI [-0.02-0.22], P+ = 0.96, BF₁₀ = 0.14, ESS_{bulk} = 4697.1, Ratio of Sig. Electrodes_{P+ > 0.95} = 38/101; OFC Estimate_{approach>return}: 0.13, 95% CI [-.01-0.27], P+ = 0.97, BF₁₀ > 0.21, ESS_{bulk} = 4350.9, Ratio of Sig. Electrodes_{P+ > 0.95} = 71/167; ACC Estimate_{approach>return}: 0.12, 95% CI [-0.07-0.31], P+ = 0.90, BF₁₀ > 0.11, ESS_{bulk} = 7437.9, Ratio of Sig. Electrodes_{P+ > 0.95} = 24/73).’

- **On lines 243-247:** ‘We did not find evidence for increased theta power in the approach period compared to the avoidance period in the MFG (Conflict-Trial Estimate_{approach>avoidance}: 0.06, 95% CI [-0.08-0.19], P+ = 0.82, BF₁₀ = 0.05, ESS_{bulk} = 4642.4, Ratio of Sig. Electrodes_{P+ > 0.95} = 31/143; Conflict-Free Estimate_{approach>return}: -0.06, 95% CI [-0.21-0.11], P+ = 0.24, BF₁₀ = 0.05, ESS_{bulk} = 4166.8, Ratio of Sig. Electrodes_{P+ > 0.95} = 11/143’.
- **On lines 347-349:** ‘We found that average connectivity values increased during the time the patient approaches (Estimate: 0.09, 95% CI [0.05-0.14], P+ > 0.99, BF₁₀ > 9, ESS_{bulk} = 7680.4, Ratio of Sig. Electrodes_{P+ > 0.95} = 1782/2816...’
- **On lines 363-365:** ‘...theta coherence drops after the patients began avoiding and moved to exit the trial (Estimate: -0.11, 95% CI [-0.16 - -0.06], P+ < 0.01, BF₁₀ > 14, ESS_{bulk} = 3520.3, Ratio of Sig. Electrodes_{P+ > 0.95} = 1824/2816...’

We additionally include the number of significant electrodes driving the directionality results in Supplemental Table 7, which we have copied here:

Region Pair	P+	Total Electrode Pairs	Total Sig. Electrode Pairs	First Region Leads (Count)	First Region Leads (Percent)
Net Granger Analysis					
MFG - > ACC	0.99	397	260	157	0.6
MFG - > OFC	0.99	552	313	218	0.7
Amyg - > OFC	0.98	321	168	104	0.62
Amyg - > ACC	0.97	65	29	27	0.93
HC - > OFC	0.94	385	170	88	0.52
HC - > ACC	0.71	90	40	25	0.62
Amyg - > MFG	0.63	95	55	25	0.45
HC - > MFG	0.61	179	63	29	0.46
OFC - > ACC	0.41	396	265	132	0.5
HC - > Amyg	0.08	336	209	107	0.51
Cross Correlation Analysis					
MFG - > ACC	0.97	397	334	189	0.57
MFG - > OFC	0.96	552	464	281	0.61
Amyg - > OFC	0.96	321	259	148	0.57
Amyg - > ACC	0.96	65	50	33	0.66
ACC - > HC	0.94	90	58	36	0.62
Amyg - > MFG	0.82	95	70	41	0.59
Amyg - > HC	0.7	336	305	128	0.42
HC - > OFC	0.46	385	273	124	0.45
ACC - > OFC	0.2	396	324	159	0.49
HC - > MFG	0.17	179	105	35	0.33

Supp. Table 7. Results for both the net Granger and cross-correlation analysis for each region pair. P+ represents the proportion of posterior samples where the effect is greater than zero, and we consider P+ > 0.95 as strong evidence for directionality. For each region pair, we also include the total number of region pairs in the dataset, the total number of pairs that showed significant directionality, the number of pairs where the first region led the second region, and the percentage of pairs where the first region led the second region out of the total number of pairs with significant directionality.

R1C12: *Figure 5F: Include amygdala data in Figure 5F for completeness, given its role in threat processing.*

Response: We thank the reviewer for this suggestion, and we have added the amygdala time-frequency plot to Figure 5f.

R1C13: *In the theta discussion, it would be interesting to consider MFG activity as reflecting conflict/danger expectation. For example, this paper [10.1038/s41467-024-54244-8] shows how MFG theta activity causally encodes conflict probability. This fits well with what was found here, especially with the high gamma activity that was the only cortical signal showing a relationship with conflict/damage probability, considering both the approach/avoidance analyses and the ghost attack analyses.*

Response: We appreciate this point and agree with the reviewer that the MFG activity could represent conflict/danger expectation and we have adapted the Discussion (beginning on line 500) to better highlight this possibility.

“Fitting with a model of approach-avoidance behavior where distal and uncertain threat results in anxiety-like behavior while proximal and certain threat induces fear behaviors (Mobbs et al., 2020; Perusini and Fanselow, 2015), we see rapid changes in the cortical-subcortical circuit during ghost attack. Specifically, the time-frequency plots were dominated by a rapid increase in high-frequency activity in the right MFG during ghost attack. Previous work has reported evidence of left/right asymmetry in the lateral PFC, including the MFG, especially in the context of emotional regulation (Brunoni et al., 2017; White et al., 2023). It has been shown that the right dorsolateral PFC is activated during periods of unpredictable threat, and this increased dlPFC activation is associated with reduced anxiety (Drevets and Raichle, 1992). These authors interpret their results as the right dlPFC enacting emotional regulation by providing top-down control on the limbic regions. In our study, the high-frequency activity in the right MFG during the acute threat period may provide a control signal as patients try to regulate their escape in the face of a near-term threat. Once patients realize they will make a successful escape the need for this control drops and is accompanied by a rapid decrease in high-frequency activity. Alternatively, or perhaps in addition to a role for emotional regulation, the activity in the MFG in this task may reflect conflict or danger expectation, as has been shown in other studies investigating approach-avoidance and other types of conflict (Abivardi et al., 2020b; Martínez-Molina et al., 2024; Zorowitz et al., 2019). MFG-limbic theta coherence increased during the approach period, and dropped during avoidance, particularly with the amygdala, and the MFG Granger causal theta oscillations in the OFC and ACC during

the approach period. It is possible that threat probability is calculated or encoded in MFG neurons and then transmitted to the OFC and ACC via theta oscillations to the OFC and ACC, which integrate that information with other signals to guide behavior. “

R1C14: Minor Edits:

- Clarify the "avoidance window" definition in control trials (line 183)
 - **Response:** We have done this, please see changes on line 230.
- Label chase/strike trials in Figure 5A for consistency.
 - **Response:** Chase and strike trials are labeled in Figure 5a.
- Check axis labels in time-frequency plots to avoid confusion (e.g., negative times = approach).
 - **Response:** We have worked to make this more clear, please see our response to comment R1C3 for a detailed explanation of the changes.

Reviewer #2 (Remarks to the Author):

Summary of the Article: In this manuscript, the authors define and investigate the role of a distributed prefrontal-limbic brain network in approach avoidance conflict. They use stereo EEG (sEEG) recorded from 20 epilepsy patients undergoing invasive monitoring for seizure localization while they performed a continuous-choice, approach-avoidance conflict decision-making task inspired by the arcade game Pac-Man. In this task, patients trade off rewards against potential losses from attack by a ghost. They show 1) Increased theta power in the hippocampus, amygdala, orbitofrontal cortex (OFC) and anterior cingulate cortex (ACC), as patients approached increasing rewards and threats, and a subsequent drop during avoidance; 2) An increase in theta band connectivity within the limbic circuit and with the medial frontal gyrus (MFG) during approach and decrease during avoidance; 3) Directional connectivity of theta from the amygdala and MFG to OFC and ACC, 4) degree of network connectivity predicted how long patients approach, with enhanced network synchronicity extending approach times and 5) when threat is imminent while being attacked by the ghost, the system dynamically switches to a sustained increase in high-frequency activity (70-150Hz) in the MFG, tracking the degree of threat. The results from this work provide evidence for a dynamic limbic-prefrontal cortical network underlying approach avoidance conflict processing in human subjects.

Comments

R2C1: Overall, this is a very well written article with clear description of methods and hypotheses being tested for each result. The findings in my opinion are significant in mapping a frontal-limbic brain circuit that is involved in approach avoidance conflict which can further be used to design interventions in patient population who have difficulty with resolving such conflict. Examples of such population as provided by the authors would be ones with anxiety disorders. However, information on such symptoms in their studied cohort of 20 patients with epilepsy is missing. A considerable population of patients with intractable epilepsy have comorbid anxiety disorders, but its not clear if any of the participants also had anxiety disorder and how that was factored in the analysis. As such, there is no information on neuropsychological characterization of the studied cohort.

Response: We thank the reviewer for their encouraging comments and for highlighting an important gap in the description of our patient sample. We have clinical information on our patient sample, which we now include in the description of the patient sample. Specifically, patients completed the 26-item Mood and Anxiety Symptom Questionnaire (MASQ-Mini) after completing the behavioral experiment. The online behavioral sample (n=122) also completed the MASQ-Mini and we break down the comparisons within this sample between online participants who reported at least one mental health disorder (n = 59) and those without (n= 63; note that we excluded our pilot sample (n=49) from these analyses, as we did not have information about any mental health diagnosis in this sample, but these data are included in the other behavioral analyses). There were three iEEG participants with elevated clinical symptoms, meaning they were above the average clinical score in the online sample that reported a mental health disorder, but otherwise symptom load was similar to the healthy sample.

Legend: Clinical symptom histograms for each subscale of the MASQ across three participant groups. The Y-axis is the number of participants and X-axis is the subscale score. The online participants are broken into two groups, those who reported having at least one mood disorder (n=59), and those who did not (n= 63; marked ‘Healthy’ in the plot). The horizontal line denotes the average subscale score for the online participants who reported at least one mood disorder. The majority (17/20) of the iEEG participants reported less symptoms than then the average symptoms in the cohort with at least one mood disorder, and the distribution of symptoms in the iEEG sample is qualitatively similar to the healthy cohort.

We examined these three patients’ neural activity in terms of their random effects from the following analyses: regional theta power elevation during the approach compared to avoidance period, increase in theta coherence during the approach period, and the correlation between approach times and theta synchrony. The three iEEG participants did not fall outside the normal distribution of the remaining intracranial cohort in any of these analyses.

We now include the above figure as Supplemental Figure 9.

R2C2: Another moderate concern is that it wasn't clear to me why the authors chose pairwise GC instead of assessing directional connectivity on the whole network such as using a partial directed coherence or directed transfer function that uses a multivariate model.

Response: We thank their attentive read of this, as it is more traditional to calculate GC on the entire network. We chose to do the GC analysis in a pairwise manner to limit the contacts to those that already demonstrated elevated theta coherence in our previous analyses. If we had assessed directional connectivity to the whole network, we would have to include edges in the network that did not have evidence of theta coherence. However, to ensure that these results were not just specific to this cohort of electrodes, we reran the Granger analysis across three additional groups with different thresholds for theta coherence. In looking at the results across the four thresholds, we see similar results with equivalent directionality. These results are also included below as a response to subsequent reviewer comment (R3C6).

Probability Positive Values by Threshold Group				
Region Pair	100ms			
	(Original)	100ms (App. Only)	500ms	1000ms
MFG -> ACC	0.994	0.993	0.991	0.988
MFG -> OFC	0.993	0.995	0.994	0.994
Amyg -> OFC	0.975	0.968	0.974	0.974
Amyg -> ACC	0.972	0.974	0.947	0.943
HC -> OFC	0.937	0.955	0.927	0.896
HC -> ACC	0.706	0.722	0.55	0.548
Amyg -> MFG	0.628	0.646	0.688	0.776
HC -> MFG	0.609	0.538	0.666	0.689
OFC -> ACC	0.412	0.321	0.341	0.323
HC -> Amyg	0.079	0.072	0.076	0.076

Legend: Probability Positive values, which describe the posterior distribution, for models estimating the directionality between region pairs, based on the Granger Causality analysis. The threshold group is defined in the following manner: '100ms' includes electrode pairs with 100ms of elevated theta coherence (corresponding to the original result), '100ms (App.)' includes electrode pairs with 100ms of elevated coherence in the approach window only, and '500ms' and '1000ms' include pairs with 500ms and 1000ms of elevated theta coherence, respectively. Grey shading indicates the Probability Positive values were greater than 0.95 in both the original thresholding and in the cross-correlation analysis.

We have worked to clarify this rationale when introducing this analysis, as reported on Lines 375-377:

"We next assessed the directionality of the prefrontal-limbic circuit using two independent methods, State-Space spectral Granger causality and cross-correlation analysis (Adhikari et al., 2010a; Barnett and Seth, 2015). We calculated the net

spectral Granger causality between all electrode pairs with significant levels of theta-band coherence. We implemented the Granger analysis in a pairwise manner to limit the electrodes to those with elevated theta coherence, as well as to better match our control analysis, which used cross-correlation (see Supplemental Tables 7-8 for estimates of results across different thresholds). As there may be bidirectional information flow, we identified which signals dominated the information flow by calculating the difference between the Granger values calculated in both directions.”

R2C3: *Some other minor concerns/comments are the following:*

- 1) *The paper title can be a bit more specific*
 - **Response:** We have updated the title to: “*Cortical-limbic circuit dynamics of approach-avoidance conflict in humans*”
- 2) *On line 286, I believe its Fig 3e and not 3g*
 - **Response:** While it was Figure 3g, we have clarified this on lines 355-358: ‘We next included an interaction term between time and region pairs to identify which region pairs were driving this effect. We found that the following pairs demonstrated strong time-coherence modulations: amygdala-MFG, amygdala-hippocampus, hippocampus-MFG, hippocampus-OFC, ACC-OFC, and ACC-MFG (See Figure 3g; Extended Data Table 1).’
- 3) *In the methods section (line 610), it wasn’t explained why the high frequency power was calculated as averages of overlapping sub bands instead of the full band.*
 - **Response:** We thank the reviewer for highlighting this missing detail. We have changed the text on lines 718-722 as follows:
 - “For the high-frequency activity estimate, we followed a similar approach but averaged the individual frequencies in 20Hz overlapping sub-bands (e.g., 70-90Hz, 80-100Hz, ... 130-150Hz) before averaging the power in each sub-band for a final estimate of high-frequency activity. We used a sub-band approach to minimize the effect of 1/f power scaling, which biases the power estimates to the lower frequencies in a large band, such as estimated from 70-150Hz.”
- 4) *The zenodo link where code will be available was missing.*
 - **Response:** We have added the necessary GitHub links (which will also be uploaded to Zenodo).
 - The link to repo for processing the neural data is here:
https://github.com/bstavel/Staveland_et_al_Pacman_Neural_Analyses
 - The link to repo for our statistical procedures are here:
https://github.com/bstavel/Staveland_et_al_Pacman_Statistics_and_Behavior

Reviewer #3 (Remarks to the Author):

Summary of the Article: The manuscript “Circuit dynamics of approach-avoidance conflict in humans” by Staveland and colleagues used intracranial EEG recordings from N=20 patients to define a prefrontal-limbic network, comprising of Amygdala, Hippocampus, OFC ACC, and MFG, that is engaged during approach-avoidance conflicts. The authors report distinct modulations of theta and HFA power during approach and avoidance, increased theta coherence during approach and decreased theta power during avoidance. The degree of network connectivity positively predicted approach time. Furthermore, high frequency activity in the MFG was observed during imminent threat. These results help to further our understanding about the neuronal basics underlying approach-avoidance in humans.

The manuscript is well written and the methods used are sound given some few clarifications. Accordingly, I have only minor comments that the authors can hopefully answer.

Comments

R3C1: *Please clarify whether the movement of Pac-Man and the ghost was continuous (i.e. as long as the subject pressed a key) or stepwise (i.e. move a certain distance every time a key was pressed)? This is not clear from the current manuscript*

Response: We thank the reviewers for identifying several aspects of the task description that needed improvement. Both the Pac-Man and the Ghost moved continuously, with a single button press initiating movement in one direction, which could reverse upon pressing the arrow key in the other direction. Based on this comment and others, we have rewritten the description of the behavioral experiment in greater detail, and the relevant changes on lines 134-135 are copied below from response to Reviewer 1.

“... The participant controls the Pac-Man and they are instructed to collect as many dots as they can while avoiding the Ghost, which is moving back and forth at a constant speed at the opposite side of the corridor. The participant makes a single button press, using the left or right arrow keys, to initiate movement along the corridor at a constant speed. By moving towards the center of the corridor, they can collect up to 5 dots, resulting in points, which varied in either small (10 points) or large (20 point) sizes and were placed in pseudorandom configurations between Pac-Man and the Ghost...”

R3C2: *The fact that the ghost was moving back and forth at the opposite end of the corridor is not mentioned in text, only in caption of figure 1. Could participants synchronize their movements of Pac-Man with those of the ghost? For instance, collect dots when the ghost moves away from the center and retreating when it returned towards the center – or would such a retreat have ended a trial?*

Response: We thank the reviewer for highlighting this important aspect of the task. We have now highlighted the Ghost’s continual movement in a section preceding the results.

Participants could synchronize their movements with the Ghost. In some participants, on some trials, participants would synchronize their movements with the Ghost, such that they would

retreat from the Ghost when the Ghost was moving towards them and then turn back when the Ghost retreated. However, there was a single approach and avoidance event per trial on most trials. The figure below shows a representative sample of online participants turning behavior (n=69). Most participants (including within the iEEG sample) only turned one time. Importantly, in cases where there were multiple turns, we always chose the last timepoint the Pac-Man turned away from the Ghost as the final avoidance decision.

Legend: Distribution of the number of turns for a subsample of online participants. The Y-axis is the trial count, and the X-axis is a subsample of participants. The stacked bar chart shows the number of trials in which there were 0 turns (participant ran into the Ghost), a single turn, 2 turns, etc. Darker colors correspond to more turns within a single trial. While most participants synchronized their movements with the Ghost on some trials, the majority of participants approached and then turned to avoid a single time.

Here is the first paragraph (beginning on line 130) of the updated task description. We have highlighted the sentences concerning the Ghost’s movement:

“**Task Design:** ... The participant controls the Pac-Man and they are instructed to collect as many dots as they can while avoiding the Ghost, which is moving back and forth at a constant speed at the opposite side of the corridor. The participant makes a single button press, using the left or right arrow keys, to initiate movement along the corridor at a constant speed. ...”

Here is the relevant portion of the updated results on lines 206-209:

“In each region, we calculated time-frequency plots time-locked to the patients’ choice to stop approaching and turn back to exit the trial. If the participant turned multiple times, we selected the final timepoint as the decision to avoid. ...”

R3C3: *In the 20% conflict-free trials, was there no ghost or did the ghost simply not attack? If the latter, how was it obvious for the subjects that these trials were conflict-free?*

Response: We appreciate the thoughtful questions about the task and have expanded details of the task design throughout the paper, and especially in a section preceding the Results.

On the conflict-free trials, there was no ghost present on the corridor. During the instructions, we explained that the PacMan always had to exit the corridor at the side it began the trial on, in to motivate similar approach and return behavior on the conflict-free trials.

Here is the relevant portion of the updated task description on lines 166-168:

'Finally, 20% of trials are conflict-free, where the participant is free to collect the 5 dots without threat of the Ghost. On these trials, the Ghost was not present on the corridor, though participants were still instructed that they had to return to exit the trial at the side of the corridor on which they began the trial.'

R3C4: *Figure 1c: I see the variability within participants but where do I see the variability across participants?*

Response: We thank the reviewer query. Each boxplot in Figure 1c is from a unique participant. The grey dots show that participant's final turning distance on each trial, and the boxplot summarizes the mean and variation within each person. By looking across the plot, it is possible to both compare which participants stayed farther from the ghost (larger means on the Y-axis), and which participants were more variable in their choices (taller boxes, more widespread dots).

However, we agree our text in the manuscript for this section was confusing. We have altered in the following way:

'Participants on average turned away from the ghost when they were 35.3+/- 3.4 units away from the ghost (corresponding to ~17% of the length of the corridor), though there was across-subject variability in how close participants were willing to get to the ghost to receive more reward (See Figure 1c).'

R3C5: *Why was no statistical test for the different turnaround distances for large and small rewards performed?*

Response: A statistical test was run to test the association between turnaround distances and large and small rewards, but the details were not explained in the manuscript and only noted in the Figure. We now have details of the model (which we updated to a Bayesian model to match the neural analyses) in three places. First, it is now described in behavioral results section on lines 181-184, which is copied below. Second, it is included in the Analysis Guide (beginning on line 684). Finally, the model output summary is included in Extended Data Table 1.

'We also hypothesized that participants would make risk/reward trade-offs tolerating more risk to get large rewards. We tested this by running a linear mixed effects model, where a two-level factor specifying if the last reward was large or small was used to predict the distance to the ghost at the time of the decision to avoid. As predicted, when the last reward in the corridor was large, participants were willing to move closer to the ghost to collect the large reward (Estimates_{small>large}: 0.11, 95% CI [0.09-0.13], $P > 0.99$, $BF_{10} > 100$, $ESS_{bulk} = 14861.4$; see Figure 1d, see Table 1 for model details; see Extended Data Table 1).'

R3C6: *Significant theta-band coherence was defined as showing at least 100ms of coherence above the shuffled distribution. Can the authors explain why precisely 100ms? Does it matter whether the 100ms were before or after the decision to avoid?*

Response: This is an important issue, as this threshold determines the group of electrode pairs carried forward to the remaining analyses. We tested how our results changed across different thresholds, including 1000ms, 500ms, 100ms within the approach window only, and without thresholding. We found that increasing the severity of the threshold generally increased the effect sizes or did not change the results. We include figures or tables demonstrating this for each analysis below, and we include each of these as a supplemental figure or table.

Coherence rises over the approach period and falls over the avoidance period: In our original analysis we found that time positively correlates with theta coherence in the approach period and negatively correlates with theta coherence during avoidance. We tested this across multiple time periods, including 1.5, 1.6, 1.7, 1.8, and 2 seconds before and after the final choice to avoid. This analysis only included electrode pairs that initially showed 100ms of elevated theta coherence across either the approach or avoidance period. Here, using the 2 seconds before and after the final choice to avoid, we show how effect strength changes when using different thresholds for theta coherence. Overall, the strength of the effect increases as we restrict the number of pairs of electrodes to those with the highest levels of theta coherence.

Legend: More stringent thresholds result in stronger estimated effects of time on theta coherence in both the approach (top) and avoidance (bottom) periods. The Y-axis is the beta estimate of the effect of time on theta coherence. The X-axis is the threshold group: ‘All Pairs’ shows the estimated effect without thresholding, ‘100ms’ shows the estimated effect with a 100ms threshold (corresponding to the original result), ‘100ms (App.)’ shows the estimated effect when thresholding by 100ms in the approach window only, and ‘500ms’ and ‘1000ms’ show the estimated effect when thresholding by 500ms and 1000ms, respectively. Error bars represent the 95% confidence intervals around the effect.

Theta oscillations in the OFC and ACC are driven by theta in the MFG and Amygdala: In our original analysis, we found that theta oscillations in the OFC and ACC were more likely to be driven by oscillations in the MFG and Amygdala than to drive them. We assessed this using two complementary approaches: pairwise Granger causality and pairwise cross-correlation analyses. Both analyses only included electrode pairs that initially showed at least 100ms of elevated theta coherence during either the approach or avoidance period.

Here, we show how P+ values change when applying different thresholds for the minimum duration of elevated theta coherence. P+ is the posterior probability that a model parameter is greater than zero. For example, in our original analysis, the MFG→ACC P+ value was 0.994, meaning that—given our data—there is a 99.4% probability that the MFG leads the ACC, and a 0.1% probability of the reverse. We interpreted results only when P+ exceeded 0.95 in both the Granger and cross-correlation analyses. Regions meeting this criterion are highlighted in grey in the table below.

Overall, the direction and statistical strength of P+ remained stable across threshold choices. The one exception was the Amygdala→ACC effect, which no longer exceeded the P+ > 0.95 criterion when requiring ≥ 500 ms or ≥ 1000 ms of elevated theta coherence, though the effect still trended in the expected direction.

Probability Positive Values by Threshold Group				
Region Pair	100ms (Original)	100ms (App. Only)	500ms	1000ms
MFG -> ACC	0.994	0.993	0.991	0.988
MFG -> OFC	0.993	0.995	0.994	0.994
Amyg -> OFC	0.975	0.968	0.974	0.974
Amyg -> ACC	0.972	0.974	0.947	0.943
HC -> OFC	0.937	0.955	0.927	0.896
HC -> ACC	0.706	0.722	0.55	0.548
Amyg -> MFG	0.628	0.646	0.688	0.776
HC -> MFG	0.609	0.538	0.666	0.689
OFC -> ACC	0.412	0.321	0.341	0.323
HC -> Amyg	0.079	0.072	0.076	0.076

Legend: Probability Positive values, which describe the posterior distribution, for models estimating the directionality between region pairs, based on the Granger Causality analysis. The threshold group is defined in the following manner: ‘100ms’ includes electrode pairs with 100ms of elevated theta coherence (corresponding to the original result), ‘100ms (App.)’ includes electrode pairs with 100ms of elevated coherence in the approach window only, and ‘500ms’ and ‘1000ms’ include pairs with 500ms and 1000ms of elevated theta coherence, respectively. Grey shading indicates the Probability Positive values were greater than 0.95 in both the original thresholding and in the cross-correlation analysis.

Theta/HFA Synchrony correlate with approach times: In our original analysis we found that synchrony between many region pairs correlated with approach times. This analysis only included electrode pairs that initially showed 100ms of elevated theta coherence across either the approach or avoidance period. Here, we show how the strength of the effect of approach times on synchrony changes when using different thresholds for theta coherence. Overall, for the theta synchrony results, the strength of the effect increases as we restrict the number of pairs of electrodes to those with the highest levels of theta coherence and are similar across thresholds for the HFA synchrony results.

Legend: Effect of different thresholds on estimated correlation between approach times and theta synchrony (top) and HFA synchrony (bottom). The Y-axis is the beta estimate of the correlation between of approach time and synchrony. The X-axis is the threshold group: ‘All Pairs’ shows the estimated effect without thresholding, ‘100ms’ shows the estimated effect with a 100ms threshold (corresponding to the original result), ‘100ms (App.)’ shows the estimated effect when thresholding by 100ms in the approach window only, and ‘500ms’ and ‘1000ms’ show the estimated effect when thresholding by 500ms and 1000ms, respectively. Error bars represent the 95% confidence intervals around the effect. Stars indicate if the region pair was significant in the original analysis.

R3C7: Regarding the Bayesian model on theta coherence (line 224): I understand that for each region one coherence value was extracted per subject and predicted by the coherence from the remaining four regions. Please provide information whether this was the average coherence over the whole-time window (-1500ms to +1500ms) or something else.

Response: We thank the reviewer for asking for more analysis details. First, we have clarified in the Results on lines 281-285 that the Bayesian model was predicting the average coherence value across the approach window, specifically including the timepoints 1500ms before the avoidance decision. Second, as discussed above, we have included an Analysis Guide (Table 1) in the Methods to help clarify details like this for each of the models in the manuscript.

Here is the relevant change of the updated results:

“We then tested if the consistent limbic theta power elevations were indicative of elevated theta coherence between limbic regions, as compared to the MFG. To test this, we ran a Bayesian mixed effects model within each region to predict the average theta coherence during the approach window (1500ms preceding the avoidance decision), using a four-level factor representing the remaining regions as the predictor.”

Here is the relevant section of the Analysis Guide (Table 1):

Finding	Dependent Variable	Fixed Effect	Random Effect	Number of Estimated Models	Number of Observations	Number of Levels	Exclusion Criteria	Figures/ Tables
Prefrontal and subcortical regions form subnetworks within a wider theta circuit	Average theta coherence during the approach period (1500ms preceding the choice to avoid) for a given region, logged	4-level factor, specifying partner region	subject/ first electrode of pair	5, one for each region	1923 - 4097	15-20/115-355	conflict-free trials; trials where 0 dots were collected	Figure 3 c; Supplemental Figures 5 & 6; Supplemental Tables 3 & 4; Extended Data Table 1

R3C8: Line 279: I assume this sentence refers to the theta power in the MFG? This is, however, not explicitly mentioned and not immediately clear.

Response: We thank the reviewer for identifying this potential point of confusion. On line 279 of the initial submission, we explain that the increase in theta coherence is unlikely to be driven by increases in theta power during the approach window because theta power is not *increasing* in the limbic regions during the approach window. However, we are not referring to theta power in the MFG, but theta power in the limbic regions, which we have clarified below.

Legend: Time-course of theta power (left subplot) and high-frequency activity (HFA, right-subplot) across approach and avoidance windows. Power data was extracted from the TFR and z-scored using the average power in the ITI. Color indicates region, shading is the standard error of the mean power across patients.

While theta power is *elevated* in the approach window compared to the avoidance window in the limbic regions, theta power in these regions begins as either *stable* or beginning to *drop* before patients make the decision to avoid. This can be seen in Figure 2d, which we have copied here.

Changes in power, especially increases in power, can sometimes artificially inflate coherence estimates. By using multiple metrics, such as imaginary coherence, and creating a permuted distribution to test for elevated coherence,

we reduce the likelihood of finding spurious increases in coherence that are driven by changes in power. Our additional point is that the increases in coherence are unlikely to be driven by increase in power, because theta power, while elevated, does not continue to increase over the approach window.

Here is the relevant change one lines 347-355 we have made to the results to clarify this:

“We found that average connectivity values increased during the time the patient approaches (Estimate: 0.09, 95% CI [0.05-0.14], $P_+ > 0.99$, $BF_{10} > 9$, $ESS_{bulk} = 7680.4$, Ratio of Sig. Electrodes $_{P_+ > 0.95} = 1782/2816$; See Figure 3d-f, Extended Data Table 1). While local theta power increases might potentially inflate connectivity estimates, this is unlikely the case because theta power – even though initially elevated – drops or remains constant at the end of the approach period, corresponding to when coherence across the network is highest (See Figure 2d). We confirmed this result using two other connectivity metrics (Pairwise Phase Consistency, Phase Lag Index) and across four different time windows ranging from 1.5-2 seconds (Supplementary Table 5).”

R3C9: For the net Granger score, 200 permutations of the trial labels were used whereas for the theta coherence 1000 permutations were used. Could the authors clarify the rationale for this discrepancy in the number of permutations? Furthermore, was the permutation procedure applied separately for each electrode pair, or was a common null distribution computed against which all net Granger values were compared?

Response: We thank the reviewer for identifying this concern. While some papers do report using 200 permutations for similar analyses, we have increased the number of permutations to 1000 and it has not changed the results.

The permutation procedure was applied individually for every electrode pair as we only wanted to include pairs with elevated theta coherence. This resulted in lengthy computation times, contributing to the original decision to have 200 permutations.

Region Pair	Prob Pos.	Total Electrode Pairs	Total Sig. Electrode Pairs	First Region Leads (Count)	First Region Leads (Percent)
MFG -> ACC	0.99	397	260	157	0.6
MFG -> OFC	0.99	552	313	218	0.7
Amyg -> OFC	0.98	321	168	104	0.62
Amyg -> ACC	0.97	65	29	27	0.93

We have changed the description of the analysis accordingly on lines 381-383:

'This final net Granger score was compared to a permuted distribution where trial labels from one electrode were shuffled 1000 times.'

R3C10: *Pairwise synchrony during approach: I understand that the correlation between envelopes of the theta signal of two electrodes was used as connectivity marker. However, the rest of the paragraph lacks sufficient detail. It is not mentioned that values were z-scored, which is only evident when looking at figure 4c. Furthermore, the authors mention Bayesian mixed model (I assume with time as predictor) but show no test statistics, only that "... coordinated theta activity within numerous region pairs in this circuit correlated with approach times." Can the authors please provide a more detailed description of the analysis and results?*

Response: We have taken three steps to make both the analysis and results more clear. First, as is mentioned above, we added an Analysis Guide to the beginning of the Methods with the important details for each of the main analyses. Second, we include a summary of the model output in the Extended Data Table. Third, we have modified the text as follows:

- **On lines 421-424:** "We tested this using Bayesian mixed effects models with random effects of patient and electrode in the set of electrode pairs that showed significantly high theta coherence. Times were log-transformed and scaled and synchrony scaled before estimating the model (see Methods and Table 1 for details)."
- **On lines 436-439:** "We used the same high theta coherence electrodes and calculated the Pearson correlation of the high-frequency activity during approach between the electrodes. Times were also log-transformed and scaled and synchrony scaled before estimating the model (see Methods and Table 1 for details)."

R3C11: *Line 348: Figure 4a-b shows the result from the Granger causality analysis, not the correlation with approach times.*

Response: We thank the reviewer for noting this and we have fixed this.

R3C12: *Colors for Amygdala and MFG in figures makes it hard to differentiate between the two.*

Response: We agree that the similar colors between the Amygdala and MFG were confusing, and we have now updated every figure so that the MFG is easier to disambiguate from the other regions.

R3C13: *Given the different sampling rates between the two hospitals: Were the iEEG data down sampled to a common sampling rate?*

Response: Thank you for identifying a missing component from our methods. Yes, the iEEG were downsampled to a common sampling rate, though this was done after we calculated power within each frequency band. We have edited the text one lines 722-744 as follows:

“Power estimates: Theta-band and high-frequency power estimates were extracted from the TFR for all statistical analyses. Power was log-transformed and each specific frequency was z-scored using an ITI baseline. Each frequency between 3-8Hz was then averaged, resulting in a theta-band power estimate. For the high-frequency activity estimate, we followed a similar approach but averaged the individual frequencies in 20Hz overlapping sub-bands (e.g., 70-90Hz, 80-100Hz, ... 130-150Hz) before averaging the power in each sub-band for a final estimate of high-frequency activity. We used a sub-band approach to minimize the effect of 1/f power scaling, which biases the power estimates to the lower frequencies in a large band, such as estimated from 70-150Hz. As the two hospitals had different sampling rates (2000Hz and 512Hz) the data were downsampled to common rate (20Hz) after power was estimated.

Christian Wienke & Marcus Grüşchow

Editor Comment: Please note that we require that the following recommendations from the guidelines are followed:

1. If the research findings apply to only one sex or gender, that must be indicated in the title and/or abstract.

Response: The findings do **not** only apply to one sex or gender.

- 2.

- a. For studies involving vertebrates animal and cell lines- The Reporting Summary should include whether sex was considered in the study design.

Response: Not applicable.

- b. For studies involving human research participants- The Reporting Summary should include whether sex and/or gender was considered in the study design and whether

sex and/or gender of participants was determined based on self-report or assigned (and methodology used).

Response: The Reporting Summary includes the following explanation of how sex and/or gender was considered in the study design:

“Reporting on sex and gender: Sex (n=20, 10 female for intracranial patients; n=191, 93 female, 3 did not report for Online behavioral participants) was self-reported and not considered in the study design due to opportunistic sampling strategy. Sex- and gender-based analyses were not performed due to the lack of statistical power in the intracranial sample. A table with patient demographics, including sex, is included in the supplement.

Recruitment: Participants in the iEEG study were recruited from all patients undergoing intracranial monitoring for clinical treatment of epilepsy during the duration of the study that met the following inclusion criteria of being over 18 years old and spoke fluent English, meaning these results may not generalize beyond these populations. Only participants with electrode coverage in at least two of the following regions were included due to prior anatomical hypotheses: amygdala, hippocampus, orbitofrontal cortex, anterior cingulate cortex, and middle frontal gyrus. Participants were informed about the possibility to participate in basic research before undergoing implantation of stereotactic EEG electrodes for the localization of seizure foci. Informed consent was obtained from those still interested in participating after implantation of the electrodes. We additionally collected 191 participants, balanced for from the online recruitment platform, Prolific, using the ‘Representative sample’ option, which balances the sample with regard to demographics like sex, based on census data from the US.”

3. Data should be reported disaggregated for sex and gender where this information has been collected and consent has been obtained for reporting and sharing individual-level data; disaggregated numbers for individual experiments must be provided in the source data as appropriate whereas overall numbers may be provided in the Nature Portfolio Reporting Summary.

Response: We now include a supplemental table with demographic information, including sex and age for each intracranial participant. This table will also be uploaded along with the source data to Zenodo. Overall numbers are reported in the manuscript and in the Nature Portfolio Reporting Summary.

4. In addition, please note that if sex- and gender-based analyses have been performed a priori, results should be reported regardless of positive or negative outcome. We discourage conducting post hoc sex- and gender-based analysis if the study design is insufficient (for example, low sample size) to enable meaningful conclusions.

Response: We do report one null report in the online behavioral sample that did not find evidence that risk/reward tradeoffs differed based on the self-reported sex of the online participants. These results are described on lines 184-187 of the manuscript:

“We did not find evidence that the self-reported sex of the participant altered how close participants were willing to get to the ghost (Estimate_{male>female}: -0.04, 95% CI [-0.16-0.08], ESS_{bulk} = 1052), nor had any interaction with reward size (Estimate_{SmallReward:Male}: 0.03, 95% CI [-0.02-0.18], ESS_{bulk} = 15356).”

Sex- and gender-based analyses were not conducted in the intracranial sample, as there was not a sufficient sample size to enable meaningful conclusions.

5. If no sex- and gender-based analyses have been performed, please indicate the reasons for the lack of these analyses in the Reporting Summary.

Response: Not applicable.